# Calculating uncertainty for the RICE ice core continuous flow analysis water isotope record

Elizabeth D. Keller[1], W. Troy Baisden[1*], Nancy A. N. Bertler[1,2], B. Daniel Emanuelsson[1,2], Silvia Canessa[1], Andy Phillips[1]

[1]National Isotope Centre, GNS Science, Lower Hutt, New Zealand
[2]Antarctic Research Centre, Victoria University of Wellington, Wellington, New Zealand
*now at: Environmental Research Institute, University of Waikato, Hamilton, New Zealand

*Correspondence to*: Elizabeth D. Keller (l.keller@gns.cri.nz)

**Abstract.** We describe a systematic approach to the calibration and uncertainty estimation of a high-resolution continuous flow analysis (CFA) water isotope ($\delta^2$H, $\delta^{18}$O) record from the Roosevelt Island Climate Evolution (RICE) Antarctic ice core. Our method establishes robust uncertainty estimates for CFA $\delta^2$H and $\delta^{18}$O measurements, comparable to those reported for discrete sample $\delta^2$H and $\delta^{18}$O analysis. Data were calibrated using a time-weighted two-point linear calibration with two standards measured both before and after continuously melting three or four meters of ice core. The error at each data point was calculated as the quadrature sum of three factors: Allan variance error, scatter over our averaging interval (error of the variance), and calibration error (error of the mean). Final mean total uncertainty for the entire record is $\delta^2$H = 0.74 ‰ and $\delta^{18}$O = 0.21 ‰. Uncertainties vary through the dataset and were exacerbated by a range of factors, which typically could not be isolated due to the requirements of the multi-instrument CFA campaign. These factors likely occurred in combination and included ice quality, ice breaks, upstream equipment failure, contamination with drill fluid, and leaks or valve degradation. We demonstrate our methodology for documenting uncertainty was effective across periods of uneven system performance and delivered a significant achievement in precision of high-resolution CFA water isotope measurement.

## 1 Introduction

Stable water isotopes ($\delta^2$H, $\delta^{18}$O) are a fundamental part of ice core studies. They are particularly important as a temperature proxy (Dansgaard, 1964; Epstein et al., 1963) and are a key component in establishing the age-depth scale and chronology of ice cores (NGRIP Members, 2004; Vinther et al., 2006; Winstrup et al., in review). They also provide other information about climate, including accumulation rates, precipitation source region, atmospheric circulation and air mass transport, and sea ice extent (e.g. Küttel et al., 2012; Sinclair et al., 2013; Steig et al., 2013; Bertler et al., 2018; Emanuelsson et al., 2018). Historically, water isotopes from ice cores were analysed as a set of discrete water samples using isotope ratio mass spectrometry (Dansgaard, 1964). Recent advances in laser absorption spectrometry have allowed continuous flow analysis (CFA) to become common in ice core studies and are an essential measurement technique for obtaining high-resolution

climate records (e.g. Kaufmann et al., 2008; Gkinis et al., 2011; Kurita et al., 2012; Emanuelsson et al., 2015; Jones et al., 2017). However, the simultaneous operation of 7 measurement systems (Winstrup et al., in review; Pyne et al., 2018) and the continuous nature of CFA poses challenges for calibration and uncertainty estimation. Because of the size and resolution of CFA ice core datasets and the relatively new application of laser spectroscopy to ice cores, few established methods exist for

calculating point-by-point uncertainty throughout measurement. Building on previous studies (e.g. Gkinis et al., 2011; Kurita et al., 2012; Emanuelsson et al., 2015), we have developed a systematic approach to calibration and error calculation that allows for unique uncertainty estimates at each data point in a CFA water isotope record. In this study, we report our methodology for the calibration and calculation of uncertainty and demonstrate the application of the method on the Roosevelt Island Climate Evolution (RICE) ice core $\delta^2$H and $\delta^{18}$O dataset.

The RICE collaboration retrieved a 760 m ice core from the north-eastern edge of the Ross Ice Shelf over Roosevelt Island in Antarctica (79.39° S, 161.46° W, 550 m a.s.l) during the austral summer 2011-12 and 2012-13 field seasons (Bertler et al., 2018). The RICE ice core provides a valuable record of a high snow accumulation site in coastal West Antarctica with annual or sub-annual resolution at the upper depths, representing the late Holocene. The climate reconstruction at the RICE site for the last 2,700 years using the CFA water isotope record is available in a separate publication (Bertler et al., 2018). In

addition to the value in the methodology itself, this manuscript provides confidence in the precision of the RICE dataset and the climatic interpretation on annual and sub-annual time scales. This method can be applied to other high-resolution CFA ice core water isotope records in the future, and may be suitable for other continuous water isotope measurement applications.

This paper is structured as follows: in Sect. 2, we give an overview of our data processing and data quality control procedure.

We also detail our methods for calibrating the isotope data and calculating the uncertainty for each data point. Section 3 contains the resulting estimates for each component of the total error of our dataset and an analysis of the different sources of error. We conclude in Sect. 4 with a summary and recommendations for future CFA measurement campaigns.

## 2 Methods

The abundance of the rare isotope in a sample is conventionally reported in delta notation, defined as:

$$\delta = \left(\frac{R_{sample}}{R_{standard}} - 1\right) * 1000 \tag{1}$$

where $R$ is $^{18}$O/$^{16}$O or $^2$H/$^1$H for water stable isotopes (Coplen, 2011). Results in this manuscript are reported as $\delta$ values in parts per thousand (‰), normalized to the international standard Vienna Standard Mean Ocean Water / Standard Light Antarctic Precipitation (VSMOW/SLAP) scale (Gonfiantini, 1978).

**2.1 Melting and data processing**

Cores were melted and processed at the Ice Core Laboratory at the GNS National Isotope Centre in Lower Hutt, New Zealand. There were two separate melting campaigns, one in June-July 2013 in which the top 500 m were melted, and the other in June-July 2014 in which the remaining 260 m (500-760 m) were melted (Pyne et al., 2018). There were several important differences between the two years in the CFA setup (Emanuelsson et al., 2015; Pyne et al., 2018), which necessitated that the data from each melting campaign be processed separately. These differences are noted where they are relevant to the calibration and uncertainty calculations; some factors were calculated individually for each melting campaign and applied only to the data from that campaign.

The ice was cut into 1 m segments and melted at a controlled rate of approximately 3 cm min$^{-1}$, producing a liquid flow rate of ~16.8 mL per minute (Pyne et al., 2018). The melting setup is based on Bigler et al. (2011) and is discussed in more detail in Emanuelsson et al. (2015), Pyne et al. (2018) and Winstrup et al. (in review). Briefly, the cores were placed vertically on a gold-coated copper melting plate and were allowed to melt continuously under gravitational pull. The water from the clean, inner part of the core was drawn from the centre of the melthead and pumped to instruments for CFA of stable isotopes, methane, black carbon, insoluble dust particles, calcium, pH, and conductivity and discrete samples for major ion and trace element analyses. The water from the outer part of the core was saved in vials for discrete stable and radioactive isotope analysis. Either three or four 1 m core segments were stacked on top of each other and melted without interruption (referred to here as a "stack"). At least one calibration cycle of three water standards was run between each stack. An optical encoder that rested on top of the core stack recorded the vertical distance displacement as the core melted. This displacement was translated into depth in millimetres, and along with the melting rate and other system information was written to a log file every 1 s using LabVIEW software (National Instruments). These log files were used to align all CFA instrument data to the depth scale. Breaks in the ice were measured and recorded to 1.0 mm precision before melting. Any ice that was cut out and removed was recorded as a gap in the depth scale. Processing of the raw data files was performed using a graphical user interface (GUI) and a semi-automated script written in Matlab (Matlab Release 2012b, The MathWorks, Inc., Natick, Massachusetts, United States). Occasionally, poor-quality ice (i.e. ice containing fractures and slanted breaks) caused the upper part of the stack to stick to the sides of the core holder; the depth encoder failed to register any change in depth for a time, while the base of the stack continued to melt. These intervals required linear interpolation (assuming a constant melt rate) and introduced a small amount of uncertainty (Pyne et al., 2018). This occurred more frequently deeper in the core in the brittle ice zone (below 500 m). Given that the melt rate was fairly constant throughout the campaign, the error introduced in the depth assignment was negligible. More details of the data processing are available in Pyne et al. (2018).

Water isotope values ($\delta^2$H, $\delta^{18}$O) were measured using CFA with a water vapour isotope analyser (WVIA) using Off-Axis Integrated Cavity Output Spectroscopy (OA-ICOS; Baer et al., 2002) and a modified Water Vapour Isotopic Standard Source (WVISS) calibration unit (manufactured by Los Gatos Research (LGR)). This system is described in detail in Emanuelsson et al. (2015). The 2013 and 2014 setups were largely the same, but differed in the construction of the vaporizer

and the delivery of the mixed vapour to the isotope analyser. In 2014, the heating element of the vaporizer was modified, and a higher sample flow was delivered directly to the IWA through an open split (Emanuelsson et al., 2015). Data was recorded in an output file at a rate of 2 Hz (0.5 s) in 2013 and at 1 Hz (1.0 s) for the remaining 260 m in 2014. The change in recording rate of the isotope data in 2014 was made to match the rate at which the depth was recorded in both years (1 Hz).

Note that this was not a change in the instrument's internal data acquisition rate, only the rate of output aggregation.

The campaigns altogether required processing and alignment of over 5 million raw data points. Depth alignment across multiple measurement systems is a key issue for ice core campaigns and a fundamental requirement for producing an age chronology (Winstrup et al., in review). The interpretation and identification of key events in the climate history thus depends on accurate depth alignment. This is particularly important deeper in the core, where a misalignment of a few

centimetres could equate to hundreds or even thousands of years (Lee et al., in review). Alignment of the isotope data to the depth scale is based on the time lag between the depth log file and the WVIA instrument output. The time lag was determined with an automated algorithm to detect the end of the calibration cycle and the beginning of the ice core melt stream using the increased change in numeric derivatives of adjacent data points. The calculated time lags during each measurement campaign averaged 418 s in 2013 and 156 s in 2014, but varied slightly from day to day by 10-20 s. (The lag

was shorter in 2014 due to the reduction in length of tubing between the melter and WVIA. Variations occurred from the periodic replacement of the tubing.) There were a few occasions of equipment failure where manual depth alignment was necessary. Poor ice quality also affected the accuracy of the depth log files, as mentioned above (Pyne et al., 2018). The precise quantification of the uncertainty introduced from the depth assignment is beyond the scope of this manuscript; based on the variation in time lags, we estimate that, at most, it is on the order of 1-10 mm.

**2.2 Data quality control**

We applied several basic selection criteria to identify and eliminate poor-quality data from the raw $\delta^2H$ and $\delta^{18}O$ dataset. The two main reasons for data removal were: 1. Changes in the water vapour concentration ($H_2O$ ppm) in the LGR analyser; and 2. The finite response time of the analyser and the transitional period when switching between water standards from the calibration cycle and RICE ice core meltwater (which by design had very different isotopic values). In addition, some gaps

were introduced as a result of cutting the core into 1 m segments and the fractures in the ice that occurred during the drilling, recovery and handling process (Pyne et al., 2018).

The isotope ratio is dependent on water vapour concentration in the analyser (Sturm and Knohl, 2010; Kurita et al., 2012). To minimize the need to correct the data for this, the concentration in the analyser was kept as close to 20,000 ppm as possible. This value was monitored and recorded at the same frequency as the isotope data. For the most part this

concentration was stable, but fluctuations and sudden changes did sometimes occur (for example, when air bubbles passed through the line). We removed data when the difference between the $H_2O$ ppm moving average over the short-term system response time of ~60 s and over a longer-term, stable time of ~200 s was greater than the standard deviation of the short-term average (Emanuelsson et al., 2015): $|avg_s - avg_l| > \sigma_s$ , where $\sigma_s$ is the standard deviation of the short-term average. In

addition, data were removed if the water vapour concentration fell below 15,000 ppm for an extended period. This filtering removed the need to further correct for variations in water vapour concentration in the record (Emanuelsson et al., 2015). Figure 1 shows a typical day of raw data, including both RICE ice core stacks and calibration cycles. Data marked in red were removed using these criteria. The majority of these points occur during the switch from one water standard to another in the calibration cycle and do not affect the data from the ice core itself. The percentage of data removed using these criteria was 0.4 % of the total.

It was also necessary to remove some data points at the beginning and end of every stack during the transition period between the Milli Q (18.2 MΩ) laboratory water standard and ice core. This transition is illustrated in Fig. 2. The Milli Q standard is composed of local de-ionised water and has an isotopic value much greater than the RICE ice core (Table 1). Milli Q was run immediately before and after each stack, and there is a period of instrumental adjustment and mixing when switching between them due to memory effects and the finite response time of the spectrometer (see Emanuelsson et al. (2015) for a full discussion). To ensure that the data is not influenced by the mixing at the beginning and end of the stack while including as much data as possible, we calculated the numerical derivative (or the rate of change) between consecutive $\delta^2H$ data points during the transition until the derivative falls below a threshold; all points prior are then excluded. The same process is performed at the end of the stack in reverse. The threshold was found empirically and is different in 2013 and 2014 because of the difference in the response times of the two setups and the precision of the data. Data was inspected manually for cases where the algorithm was inadequate. Approximately 2-5 cm of the beginning and end of every stack were removed using this condition. These appear as gaps in the depth of the final dataset. There were also a few occasions when melting was interrupted due to equipment failure, and Milli Q was run through the system until melting could resume; these periods were removed using the same procedure. A typical stack showing a portion of data removed is shown in Fig. 2 ($\delta^2H$ vs. depth). The fraction of total data removed was 5.4 %. This resulted in short data gaps of 2-5 cm every three or four meters.

The entire dataset was manually inspected for any other regions of poor quality, and points that visibly fell outside the normal range or were affected by known instrument problems were removed. This only applied to a few isolated sections of data and was a very small portion (< 0.1 %) of the total.

### 2.3 Calibration

It is necessary in laser spectroscopy to normalize the isotopic values to the VSMOW/SLAP scale and to correct for instrumental drift. To accomplish this, we used a two-point linear calibration method (Paul et al., 2007; Kurita et al., 2012). Before and after each ice core stack, we ran calibration sequences consisting of four laboratory water standards: Milli Q, Working Standard 1 (WS1), RICE snow (RICE), and US-International Trans-Antarctic Scientific Expedition West Antarctic snow (ITASE). An example of a calibration cycle is shown in Fig. 3. Assigned or "true" values for these standards as measured against the VSMOW-2/SLAP-2 scale are listed in Table 1. Each batch of working standards was calibrated to the International Atomic Energy Agency (IAEA) primary standards VSMOW-2 ($\delta^{18}O = 0.0$ ‰; $\delta^2H = 0.0$ ‰), SLAP-2 ($\delta^{18}O = -$

55.50 ‰; $\delta^2H$ = -427.5 ‰), and GISP ($\delta^{18}O$ = -24.76 ‰; $\delta^2H$ = 189.5 ‰) using three intermediate, secondary standards INS11 ($\delta^{18}O$ = -0.37 ‰; $\delta^2H$ = -4.2 ‰), CM1 ($\delta^{18}O$ = -16.91 ‰; $\delta^2H$ = -129.51 ‰) and SM1 ($\delta^{18}O$ = -28.79 ‰; $\delta^2H$ = -225.4 ‰).

We note that there is a difference in the assigned values for RICE and ITASE between 2013 and 2014. We have denoted
them RICE-13, RICE-14, ITASE-13 and ITASE-14 in Table 1 to indicate that these standards were prepared and stored in different batches in each year, from water sources that had not been treated as standards or homogenized, and thus are slightly different in composition. We emphasize here that our standards are local working standards, selected or mixed by our laboratory to match the isotope ratios of the sample (melt stream). It is not unexpected that their isotopic value will change between batches during long measurement campaigns, as it is not practical to prepare and store all of the material in
one batch.

Part of the difference in assigned values might be attributed to the difference in measurement systems. The assigned values for the 2013 calibrations were determined using discrete laser absorption spectroscopy measurements on an Isotope Water Analyzer (IWA) 35EP system. In 2014, our instrument was upgraded with a second laser to IWA-45EP, and the 2014 calibrations utilize values from standards measured continuously with this system. We were regrettably not able to calibrate
our working standards using the 2013 CFA setup before the setup was modified for the 2014 campaign, so we use the assigned values from the 2013 discrete measurements in the 2013 calibrations. We thus consider the 2014 melting campaign to be better calibrated than the 2013 campaign. This follows from the principle of "identical treatment" (IT) of stable isotope analysis wherein samples and reference materials should be subject to identical preparation, measurement pathways and data processing to the extent possible (Werner and Brand, 2001; Carter and Fry, 2013; Meier-Augenstein, 2017).

The working standards used for the calibration, RICE and ITASE, have assigned values which form an upper and lower bound, respectively, for the majority of the ice core isotopic values (the ice core samples from the younger, top portion of the core occasionally fall slightly above the RICE standard). The third water standard (WS1) served as a quality control to enable us to check and quantify the accuracy of the calibration. Each standard was run continuously for approximately 10 minutes (varying between 8-15 min over the course of the melting campaigns), of which the first and last 100-200 s were
discarded to ensure only the middle, stable portion of the measurement was used for calibrations. Around 300 s of data were averaged to arrive at the mean value of the measurement.

Frequent measurements of calibration standards are necessary to correct isotopic measurements for instrumental drift over time. At least one cycle of all three standards was run between stacks, and in many cases, there were several cycles. Melting a stack of three or four cores took around 2-2.5 hours, so the measurement at the mid-point of a stack (the points furthest
from a calibration) is about 1-1.25 hours from the nearest calibration. While this is longer than would be ideal for isotope laser spectroscopy, the stability of other elements of the CFA system (in particular, continuous flow methane measurements) required long uninterrupted periods of melting. $\delta^{18}O$ is typically more affected by drift than is $\delta^2H$. Drift can be worsened by experimental conditions such as drill fluid contamination and leaks in the system as the analyte proceeds toward the vacuum

in the laser cavity. We have quantified the error introduced by the amount of drift occurring between calibrations using the Allan deviation, discussed in Sect. 2.4.1.

We have used a two-point linear normalisation procedure, which is routinely used to adjust measured δ-values to an isotopic reference scale (Paul et al., 2007). The correction takes the form of linear regression: $\delta_{corrected} = m * \delta_{measured} + b$ ,

where $m$ is the slope of the line and $b$ is the y-intercept. The measured δ-values of two laboratory standards are regressed against their assigned δ-values. The slope $m$ can be calculated by plotting the measured values of the standards on the x-axis and their assigned values on the y-axis and then using trigonometric formulas to relate them to the true value of the sample (Paul et al., 2007). The result is the ratio of the difference between the true RICE and ITASE δ values and the actual difference measured:

$$m_i = \frac{\delta^T_{RICE} - \delta^T_{ITASE}}{\delta_{RICEi} - \delta_{ITASEi}}$$            ( 2 )

where $\delta^T_{RICE}$ and $\delta^T_{ITASE}$ are the assigned "true" values and $\delta_{RICEi}$ and $\delta_{ITASEi}$ are the $ith$ measured values of the standards RICE and ITASE, respectively. The correction then takes the following form:

$$\delta_{corrected} = \frac{\delta^T_{RICE} - \delta^T_{ITASE}}{\delta_{RICEi} - \delta_{ITASEi}} * (\delta_{raw} - \delta_{RICEi}) + \delta^T_{RICE}$$            ( 3 )

By design, the y-intercept or offset $b$ is equal to the difference between $\delta^T_{RICE}$ and $\delta_{RICEi}$ when the slope $m$ is 1. We applied

this correction to each data point by weighting the factors calculated from the RICE and ITASE calibration measurements both before and after the stack with the time difference between the data point and the calibration:

$$\delta_{corrected}(t) = \left[(\delta_{raw} - \delta_{RICE1}) * m_1 + \delta^T_{RICE}\right] * (1 - f) + \left[(\delta_{raw} - \delta_{RICE2}) * m_2 + \delta^T_{RICE}\right] * f$$       ( 4 )

where $\delta_{raw}$ is the uncalibrated, raw δ²H or δ¹⁸O value of the ice core sample, $\delta_{RICE1}$ and $\delta_{RICE2}$ are the measured values of the RICE standard before and after the stack, respectively, $t$ is the time of the $\delta_{raw}$ measurement, and $f$ is a dimensionless

weighting factor: $f = (t - t_1)/(t_2 - t_1)$, $t_1$ = starting time of $\delta_{RICE1}$ measurement before the stack, and $t_2$ = ending time of $\delta_{RICE2}$ measurement after the stack. We note that this method assumes that drift is approximately linear over the measurement period. Our calibration procedure was validated by comparison to discrete measurements in Emanuelsson et al. (2015). The values of the slope corrections and the RICE and ITASE raw measurements used to calibrate the data in each year are shown in Figs. S2-S4; mean values and standard deviations are in Table S1.

**2.4 Uncertainty calculation**

We identified three main sources of uncertainty in our measurements: (i) the Allan variance error (a measure of our ability to correct for drift, a systematic source of uncertainty due to instrumental instability), (ii) the scatter or noise in the data over our chosen averaging interval, and (iii) a general calibration error relating to the overall accuracy of our calibration. Our three error factors can be formally categorized as follows:

1.   "Allan variance error": systematic error or bias due to our imperfect ability to correct for drift

      2.   "Scatter error": error of the variance / precision / random variation of replicate measurements

3. "Calibration error": error of the mean / trueness

The last two can be quantified with general analytical expressions (Kirchner, 2001). Systematic error or bias does not have a general analytical form; isotopic drift is fortunately amenable to correction, but the method is imperfect.

We assume that the three error factors are uncorrelated to a large degree. This is supported by the general framework that we have used (Kirchner, 2001; Analytical Methods Committee, 2003) and the actual errors calculated at each data point ($R^2 <$ 0.05 in each year for both isotopes). In practice it is impossible for all error factors to be completely uncorrelated, as some underlying sources of error will affect all aspects of the system. However, our data suggest that these interactions are small and/or short-lived and negligible to the total uncertainty. With this assumption, we calculate each error factor separately and add them in quadrature to arrive at the total uncertainty estimate:

$$\sigma_{total} = \sqrt{\sigma_{AVE}^2 + \sigma_{scatter}^2 + \sigma_{calib}^2} \qquad (5)$$

Each data point in the final record is assigned a unique error value. A detailed explanation of the calculation of each source of uncertainty follows.

### 2.4.1 Allan variance error

The Allan variance $\sigma_{allan}^2$, or two-sample frequency variance (Allan, 1966), is often used as a measure of signal stability and instrumental precision in laser spectroscopy (Werle, 2011; Aemisegger et al., 2012). In the context of CFA isotope measurements, it is also used as an estimate of how much instrumental drift accumulates over a specified period. It is defined by:

$$\sigma_{allan}^2(\tau) = \frac{1}{2n}\sum_{j=1}^{n}\left(\delta(\tau)_{j+1} - \delta(\tau)_j\right)^2 \qquad (6)$$

where $\tau$ is the averaging time, n is the number of time intervals, and $\delta(\tau)_j$ and $\delta(\tau)_{j+1}$ are the mean values of adjacent time intervals $j$ and $j+1$ with length $\tau$. The Allan deviation is the square root of the variance, $\sigma_{allan}$.

We calculated the Allan deviation of our system using measurements of the Milli Q standard, run continuously for 24-48 hours. We conducted these tests periodically during both measurement campaigns (usually over the weekend when the instruments were otherwise idle; see Emanuelsson et al., (2015) for details). On a log-log plot of the Allan deviation vs. averaging time ($\tau$), there is a minimum at the averaging time where the precision is highest; before this point, at very short averaging times, instrumental noise affects the signal, and after, at longer averaging times, the effects of instrumental drift can be seen. Thus, the Allan deviation provides an estimate of the optimal averaging time, before and after which precision decreases.

The Allan deviation can also provide an indication of the uncertainty due to instrumental drift as a function of the time difference between the measurement and the nearest calibration. For our system to stay under the precision limit of 1.0 ‰ and 0.1 ‰ for $\delta^2H$ and $\delta^{18}O$, respectively (and to permit analysis with deuterium excess, d = $\delta^2H$ - 8 * $\delta^{18}O$), a calibration cycle to correct for drift should occur at least every ~1 hr during ice core measurements (Emanuelsson et al., 2015).

However, as noted above, system limitations prevented us from running calibrations as frequently as would have been optimal. We use the Allan deviation here to estimate how quickly instrumental drift increased and thus how well we were able to correct for drift using our calibrations.

We plot the mean $\sigma_{allan}$ for all tests performed against averaging time $\tau$ on a log-log scale (done separately for 2013 and 2014) and perform a linear regression on the curve for averaging times greater than the minimum $\sigma_{allan}$. The equation of the linear fit gives what we refer to as the "Allan variance error" (denoted $\sigma_{AVE}$ to distinguish our error from the official definition of the Allan deviation):

$$\log_e \sigma_{AVE} = a * \log_e t + b \tag{7}$$

$$\sigma_{AVE} = t^a * e^b \tag{8}$$

where $t$ is the time difference between the data point and the calibration (as measured from the start of the measurement of the RICE standard), and $a$ and $b$ are constants determined from the linear regression. This error factor is calculated for each data point as a function of $t$. Because we calibrated using standards measured both before and after each stack, there are two factors at each point that are combined with a time-weighted average, using the same weighting used for the calibration (Eq. 4):

$$\sigma_{AVE}(t) = \left( |t - t_1|^a * e^b \right) * (1 - f) + \left( |t - t_2|^a * e^b \right) * f \tag{9}$$

where $f$ is defined as before in Sect. 2.3. Allan variance error vs. depth over the whole dataset is shown in Fig. 4. The local maximum for each stack occurs in the middle, at the point furthest away in time from the two calibrations bracketing the stack, reflecting that it is at this point that we are most uncertain of the amount of instrumental drift.

### 2.4.2 Scatter error

A second error derives from the scatter or noise in the signal over our averaging interval (15 s). This averaging interval was chosen by the RICE project team as a suitable scale to smooth over measurement noise without obscuring important features in the data. This equates to approximately 7-8 mm on the depth scale. Due to this deliberate choice, the error calculation that follows applies over this interval. To quantify this analytical uncertainty, we calculate the standard deviation for every 15-s time interval contained in each measurement of the RICE standard using a moving window (so that each adjacent, overlapping interval is advanced by 1 s) and average over the duration of the measurement:

$$\sigma_{scatter} = \frac{1}{N} \sum_1^N \frac{\sigma_i}{\sqrt{n_i}} = mean\left( \sigma_i / \sqrt{n_i} \right), \quad i = [1 \dots N] \tag{10}$$

where $\sigma_i$ is the standard deviation, $N$ is the total number of intervals, and $n_i$ is the number of data points in the $ith$ interval ($n = \sim30$ in 2013 and $\sim15$ in 2014). We note that the number of points that are contained in the interval is different in 2013 and 2014, resulting from the difference in output aggregation (not the instrument's internal data acquisition rate). This could affect the amount of noise in the data. However, we have not attempted to analyse this in detail, as we are only concerned

here with quantifying the uncertainty associated with our averaging interval, regardless of the number of data points averaged.

Again, because the RICE standard was measured both before and after each stack, we calculate $\sigma_{scatter}$ for both measurements and linearly combine them using a time-weighted average. Note this error is linear with time within a stack but is discontinuous at the points where a stack begins and ends. This linearity is rooted in the fact that the noise in a set of measurements from the same sample can in general be modelled as a Gaussian process, with a normal distribution of independent random variables. The mean-squared displacement is linear with time. Scatter error vs. depth for the length of the core is shown in Fig. 5.

### 2.4.3 Calibration error

Finally, we calculate the error of the mean after applying our calibration procedure as a measure of the trueness of the measurement with respect to our reference scale, denoted by $\sigma_{calib}$. This captures both random, unsystematic components of uncertainty and systematic biases in the calibration stemming from a variety of (unspecified) sources. This measure is often taken as a check on the overall quality of the calibration procedure. Because it encompasses multiple sources of error, we expect it to be a relatively large error. Here we make use of the large set of WS1 measurements that were made during the calibration cycles. To calculate this factor, we apply the calibration formula using the RICE and ITASE standards (Eqs. (2) and (3)) to the third quality-control standard, WS1, measured in the same cycle. The error is defined as the difference between the corrected, measured value and the assigned value of the WS1 standard. An example is shown in Fig. 6. We calculated this difference for all calibration cycles containing measurements of all three standards (RICE, ITASE and WS1) of sufficient quality (there were 221 such calibration cycles in 2013 and 318 in 2014) and then took the mean of the differences. Separate error estimates for the 2013 and 2014 melting campaigns were calculated and applied only to the data points from the respective year. The calibrated values obtained for all of the WS1 measurements throughout both campaigns are show in Fig. S1.

### 3 Results and Discussion

Total error vs. depth for the whole record is shown in Fig. 7 and summarized in Table 2. The mean total error for all data points is 0.74 ‰ ($\delta^2$H) and 0.21 ‰ ($\delta^{18}$O). Separated by melting campaign, mean total error in 2013 is 0.85 ‰ ($\delta^2$H) and 0.22 ‰ ($\delta^{18}$O) and in 2014 is 0.44 ‰ ($\delta^2$H) and 0.19 ‰ ($\delta^{18}$O). The total error reduces sharply at a depth of 500 m due to the switch between 2013 and 2014 campaigns and the greatly reduced calibration error in 2014. However, we observe a larger variability in the error in the 2014 data. This is mainly a result of the highly variable amount of noise in the measurements, which is discussed below.

The mean Allan error for all data is 0.12 ‰ for $\delta^2$H and 0.14 ‰ for $\delta^{18}$O. Calculated separately by melting campaign, the mean errors are 0.13 ‰ ($\delta^2$H) and 0.16 ‰ ($\delta^{18}$O) in 2013 and 0.083 ‰ ($\delta^2$H) and 0.11 ‰ ($\delta^{18}$O) in 2014. As expected, the

Allan error peaks at the points in the middle of the stack furthest from a calibration (Fig. 4). It is both absolutely and proportionally larger for $\delta^{18}$O, as $\delta^{18}$O is typically more affected by drift.

The amount of scatter in the data varies considerably over the length of the record, particularly in 2014. The mean scatter error over the whole record is 0.29 ‰ ($\delta^2$H) and 0.10 ‰ ($\delta^{18}$O). Separated by melting campaign, the mean errors are 0.26 ‰ ($\delta^2$H) and 0.093 ‰ ($\delta^{18}$O) in 2013, and 0.37 ‰ ($\delta^2$H) and 0.13 ‰ ($\delta^{18}$O) in 2014. On average, the scatter error is larger in 2014, although during the periods of best instrumental performance, $\sigma_{scatter}$ is lower than at any point in 2013. The instrument performance was highly variable in 2014, much more so than 2013. The standard deviation of $\sigma_{scatter}$ is 0.11 ‰ ($\delta^2$H) and 0.045 ‰ ($\delta^{18}$O) in 2014, as opposed to 0.026 ‰ ($\delta^2$H) and 0.012 ‰ ($\delta^{18}$O) in 2013.

Among the three error factors, the general calibration error is the largest contributor to the total error in 2013: $\sigma_{calib}$ ($\delta^2$H) = 0.80 ‰ and $\sigma_{calib}$ ($\delta^{18}$O) = 0.12 ‰. However, this error is greatly reduced for 2014: $\sigma_{calib}$ ($\delta^2$H) = 0.22 ‰ and $\sigma_{calib}$ ($\delta^{18}$O) = 0.078 ‰, reflecting the improved measurement of the assigned standard values. We were not able to measure the standards against VSMOW/SLAP using the 2013 CFA setup (time constraints did not permit us to conduct additional measurements after the 2013 campaign concluded, as our instrument was sent to the manufacturer for modification), which would provide a better comparison between measured and assigned values, following from the principle of identical treatment (Werner and Brand 2001). The 2013 $\sigma_{calib}$ is thus likely to be a very conservative estimate of the error. In addition, the assigned value of WS1 is well outside the range of the RICE ice core and is much greater than the RICE and ITASE standards, and thus RICE and ITASE could be considered a poor choice for calibrating WS1. The two calibration standards, RICE and ITASE, were chosen to be similar in isotopic value to the ice core samples being measured (Werner and Brand 2001), with the quality-control standard being of secondary concern. Ideally, we would use a quality-control standard that falls within the range of the values of our two calibration standards. While we could have used WS1 and ITASE as our calibration standards and RICE as a quality-control, WS1 is less appropriate than RICE for calibrating the range of isotopic values found in the ice core. Testing the sensitivity of the calibration error to our selection of quality-control standards, however, is outside the scope of this manuscript.

The scatter error dominates the total error in 2014. The magnitude of this error was highly variable from day to day, and thus the total error also varied considerably. There were some periods in which the instrument performed exceptionally well. During these periods, total error was as low as 0.3 ‰ ($\delta^2$H) and 0.1 ‰ ($\delta^{18}$O). These represent the high end of system capability. However, for much of the 2014 melting campaign the total error was closer to the average of 0.44 ‰ ($\delta^2$H) and 0.19 ‰ ($\delta^{18}$O).

There are three main possible reasons for the large variations in performance in 2014. They are: 1) response to breaks in the ice and associated bubbles; 2) performance degradation due to unexpected levels of drill fluid in the melt stream (a mixture of Estisol-240 and Coasol was used to keep the drill hole open; although all pieces of ice were thoroughly cleaned before melting, some contamination occurred through existing microfractures in the ice); 3) leaks or valve degradation in the laser spectrometer, which operates under vacuum. There were significantly more performance issues in 2014. In addition to the different setup and gradual build-up of drill fluid in the instruments over time, the ice itself was of poorer quality at deeper

depths (especially in the brittle ice zone at depths below 500 m; Pyne et al., 2018), containing more breaks that caused interruptions in the CFA measurements and possible drill fluid contamination. Although we have only anecdotal evidence, the more frequent stopping and restarting of the system in 2014 seemed to introduce more noise into the measurements. Because the campaign was conducted to operate many measurement systems simultaneously, as is characteristic of ice core

CFA campaigns, it was typically not possible to conduct comprehensive performance tests and systematic evaluations during the one day of down time in each week-long, seven-day cycle. As a result, the precise sources of performance deterioration were difficult to isolate. Our method for calculating uncertainty is designed to capture the changing day-to-day conditions resulting from a range of system variations and performance issues, even if it is not possible to pinpoint the exact cause.

## 4 Summary and conclusions

We have described a systematic approach to the data processing and calibration for the RICE CFA stable water isotope dataset and presented a novel methodology to calculate uncertainty estimates for each data point derived from three factors: Allan deviation, scatter, and calibration accuracy. The mean total error for all data points is 0.74 ‰ ($\delta^2$H) and 0.21 ‰ ($\delta^{18}$O). Mean total error in 2013 is 0.85 ‰ ($\delta^2$H) and 0.22 ‰ ($\delta^{18}$O) and in 2014 is 0.44 ‰ ($\delta^2$H) and 0.19 ‰ ($\delta^{18}$O). This represents a significant achievement in precision of high-resolution CFA water isotope measurement, and documentation of uncertainty

calculations for isotope analyses in a continuous measurement campaign comprising multiple complex measurement systems.

The isotope analyser system performed exceptionally well during some time intervals in 2014, demonstrating high capability, even though this was not sustained. The variability in quality could be due to poor ice quality, interruptions in the CFA measurements, the build-up of residual drill fluid in the instrument, and / or leaks and valve degradation. Most likely it

was a combination of all of these factors.

The more accurate measurement of our laboratory water standards for the 2014 melting campaign enabled us to reduce the uncertainty considerably for the data at depths greater than 500 m. More generally, a reduction in the uncertainty in the system could be achieved through more rapid calibration cycles, enabling both the insertion of calibration during "stacks" and more rapid troubleshooting to isolate causes of degraded performance.

Our uncertainty estimates do not take into account the additional uncertainty introduced from the smoothing of the data during the melting procedure and the measurement response time. This is an important issue, particularly for deep, older ice, where annual layers are greatly compressed and measurement resolution is crucial to the ability to date the core accurately. The degree of mixing in the melting procedure itself can be controlled through the melting rate and the diameter of tubing leading from the melter to the CFA instruments. Our system was designed primarily for high throughput and multiple,

simultaneous measurements. However, these parameters can be adjusted to increase resolution for older ice (the very bottom of the RICE core has yet to be measured).

The volume of the evaporation chamber is usually a limiting factor in the temporal resolution and response time of the IWA and can introduce a significant amount of uncertainty. While we reduced the volume of the chamber from the manufacturer's default of 1.1 L to 40 mL (Emanuelsson et al., 2015), there is still a finite time required to fill and replace the chamber with new sample. We estimate that our depth resolution was between 1.0 – 3.0 cm (Pyne et al., 2018). A more comprehensive evaluation of the effect of the mixing inherent in the melting and measurement procedure on the overall uncertainty is beyond the scope of this paper but is an important consideration for future work.

**Acknowledgements**

Funding for this project was provided by the New Zealand Ministry of Business, Innovation, and Employment Grants through Victoria University of Wellington (RDF-VUW-1103, 15-VUW-131) and GNS Science (540GCT32, 540GCT12), and Antarctica New Zealand (K049). We are indebted to everyone from the 2013 and 2014 RICE core processing teams. We would like to thank the Mechanical and Electronic Workshops of GNS Science for technical support during the RICE core progressing campaigns. This work is a contribution to the Roosevelt Island Climate Evolution (RICE) Program, funded by national contributions from New Zealand, Australia, Denmark, Germany, Italy, China, Sweden, UK and USA. The main logistic support was provided by Antarctica New Zealand and the US Antarctic Program.

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

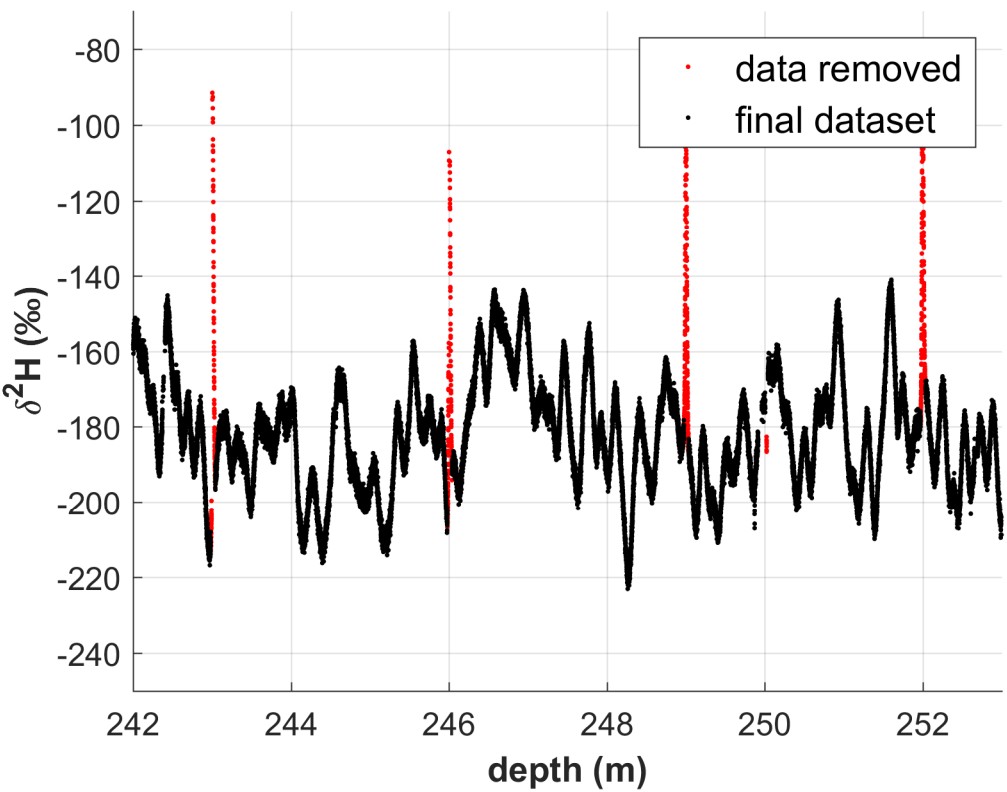

**Figure 2: A selected example section of δ²H vs. depth. The data marked in red represent the transitions between the Milli Q standard and ice core at the boundaries of each 3-metre stack. These data points (and other poor quality data) were removed from the final dataset.**

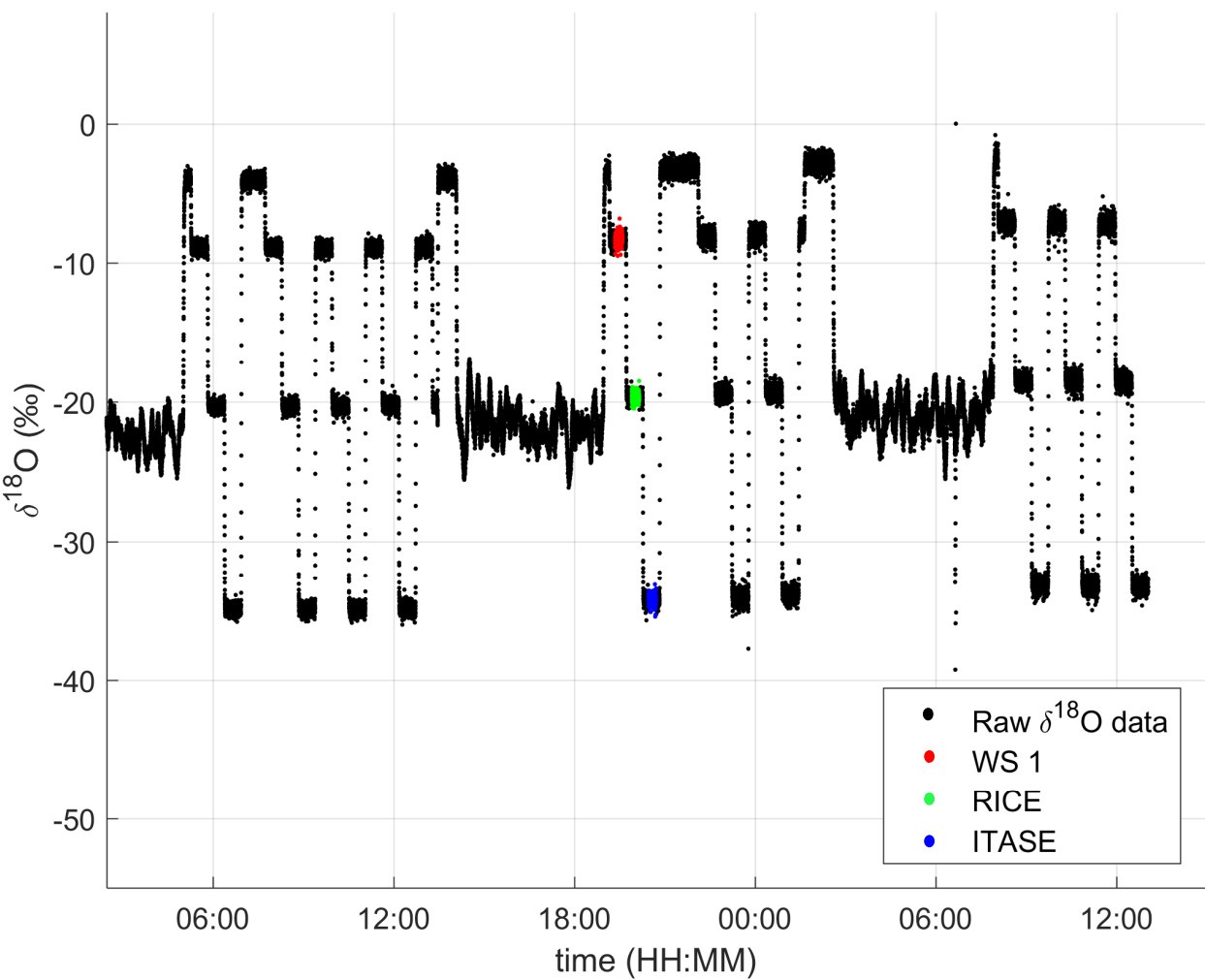

**Figure 3: Time vs. raw δ¹⁸O (uncalibrated) for one day of melting (3 July 2014). Values of standards drift noticeably over the course of the day. An example of one calibration cycle of three water standards run between ice core stacks is marked in colour: WS1 (red), RICE (green), and ITASE (blue).**

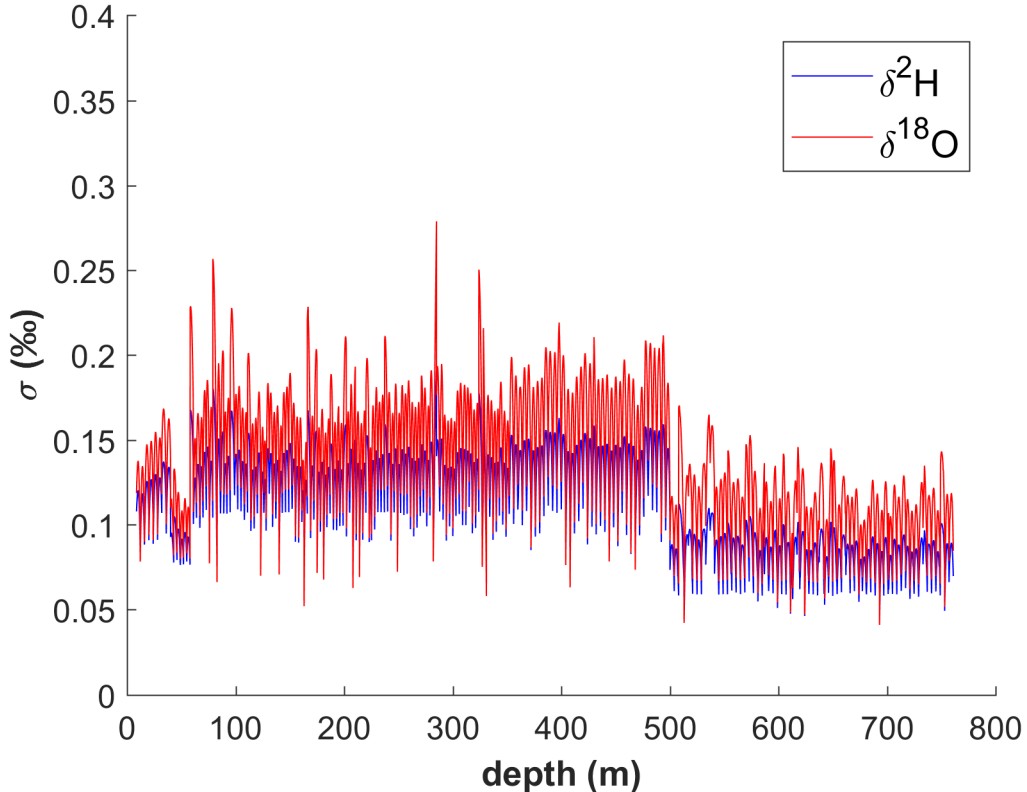

**Figure 4: Allan variance error vs. depth, in per mil. δ²H is in blue and δ¹⁸O is in red. The low points of the dips are the start and end of a stack, between which calibrations were carried out.**

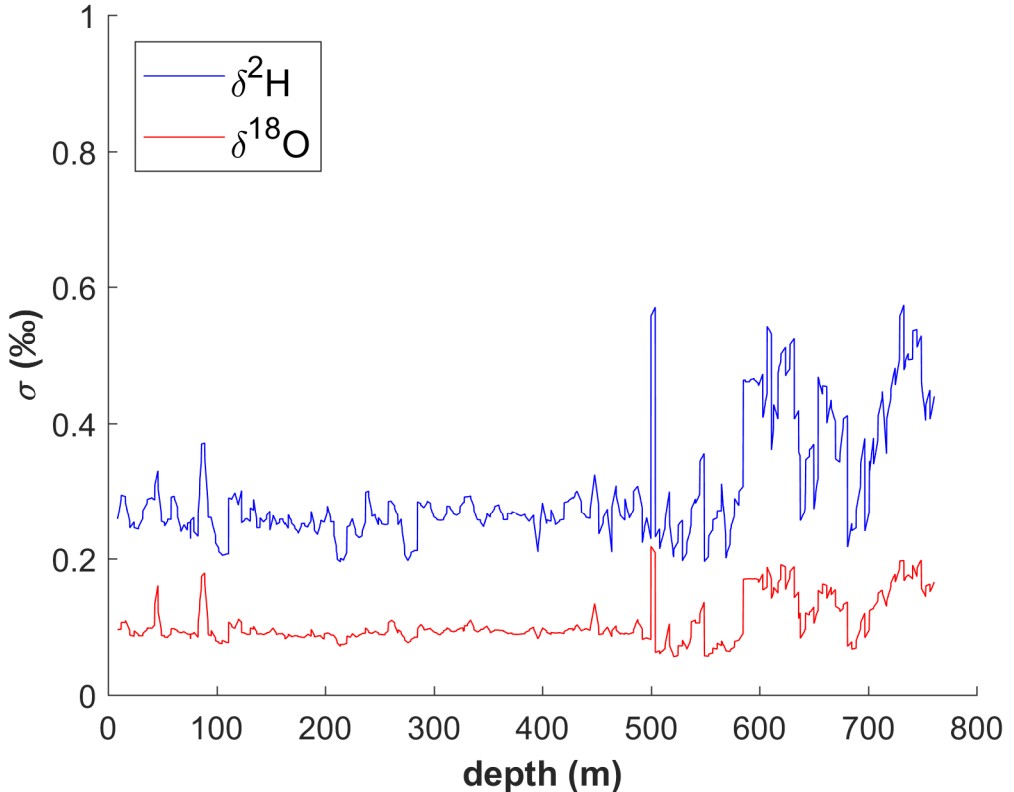

**Figure 5: Scatter error vs. depth, in per mil. δ²H is in blue and δ¹⁸O is in red.**

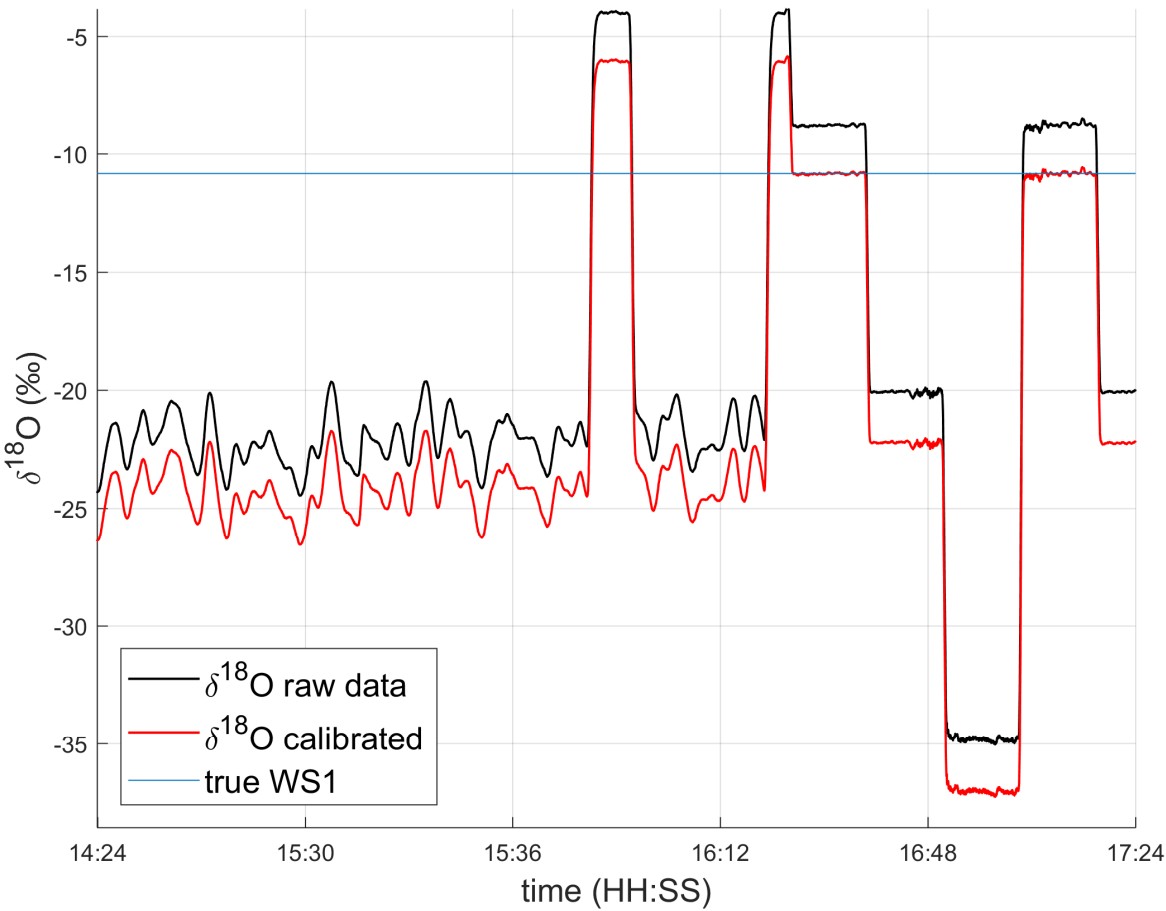

**Figure 6: Representative δ¹⁸O calibration of ice core stack and WS1, using RICE and ITASE standards from the same cycle, 15-second moving average vs. time (measured on 2 Jul 2014). The difference between the "true" value of WS1 (blue) and the calibrated measured value of WS1 (red) is the calibration error. The error that was applied to the CFA dataset is the average difference of all WS1 calibration measurements during the melting campaign.**

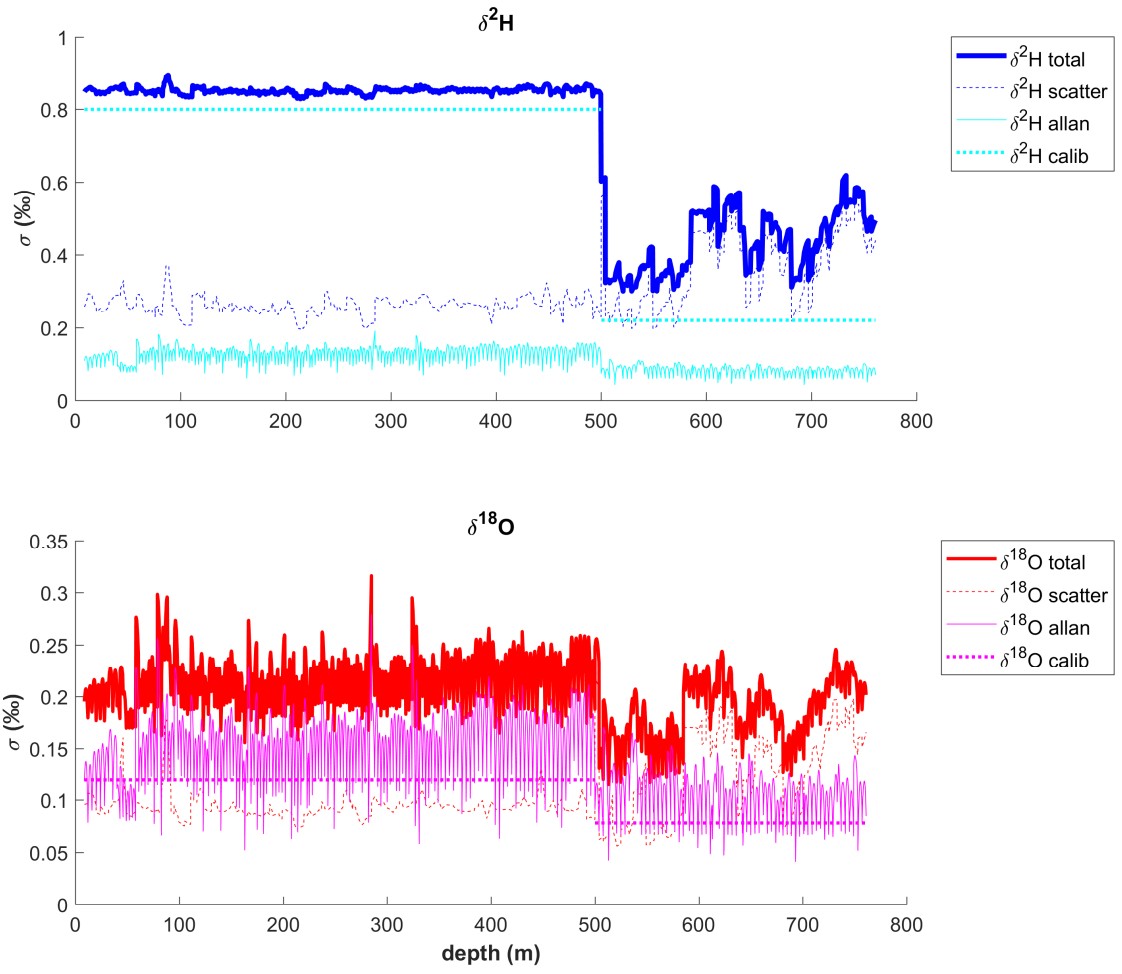

Figure 7: Total uncertainty vs. depth, along with each individual error factor, in per mil. Top: δ²H. Bottom: δ¹⁸O. There is a noticeable discontinuity at 500 m; the melting campaign was paused at 500 m in 2013, and melting was resumed in 2014 with a modified setup. The reduced calibration error in 2014 is responsible for the large step down in total error.

**Tables**

**Table 1: Accepted values (VSMOW/SLAP scale) for water standards used for calibrations, in per mil (‰).**

| Water standard | $\delta^{18}O$ (‰) | $\delta^2H$ (‰) |
|---|---|---|
| Milli Q | -5.89 +/- 0.05 | -34.85 +/- 0.18 |
| WS1-13 | -10.84 +/- 0.10 | -74.15 +/- 0.94 |
| WS1-14 | -10.83 +/- 0.05 | -74.85 +/- 0.18 |
| RICE-13 | -22.54 +/- 0.05 | -175.02 +/- 0.19 |
| RICE-14 | -22.27 +/- 0.05 | -173.06 +/- 0.24 |
| ITASE-13 | -37.39 +/- 0.05 | -299.66 +/- 0.18 |
| ITASE-14 | -36.91 +/- 0.08 | -295.49 +/- 0.52 |

**Table 2: Summary of error estimates, in per mil (‰).**

| | $\delta^{18}O$ (‰) | | | $\delta^2H$ (‰) | | |
|---|---|---|---|---|---|---|
| **Error factor** | **2013** | **2014** | **Combined** | **2013** | **2014** | **Combined** |
| Allan | +/- 0.16 | +/- 0.11 | +/- 0.14 | +/- 0.13 | +/- 0.083 | +/- 0.12 |
| Scatter | +/- 0.093 | +/- 0.13 | +/- 0.10 | +/- 0.26 | +/- 0.37 | +/- 0.29 |
| Calibration | +/- 0.12 | +/- 0.078 | n/a | +/- 0.80 | +/- 0.22 | n/a |
| **Total** | +/- **0.22** | +/- **0.19** | +/- **0.21** | +/- **0.85** | +/- **0.43** | +/- **0.76** |

