# Peer review of "Calculating uncertainty for the RICE ice core continuous flow analysis water isotope record"

_Atmospheric Measurement Techniques, 2017_

## Referee Comment (RC1) · Anonymous Referee #1 · 13 Mar 2018

**1   Overview**

The manuscript by Keller et al. presents methods to assess the uncertainty of the water isotope ratio measurement for an ice core continuous flow analysis (CFA) system used for the measurements of the RICE ice core. This is a work that builds on previously published methods by Emanuelsson et al. (2015). The work focuses on a rather special but very essential part of CFA system for water isotopic analysis that of the uncertainty characterisation. It fits very well within the scope of the Atmospheric Measurements Techniques journal. I find it very positive that the authors decide to focus a separate manuscript for assessing the uncertainty of the water isotopic measurement. This is not a usual practice and it is most welcome.

[Figure]

Unfortunately though i cannot recommend the manuscript for publication. I would like to let the decision to the editor (based also on the assessement of the other reviewers) on weather the authors should proceed with a major revision of the manuscript or withdraw their submission and start fresh. The second option would actually be my recomendation. The reason for this decision is that almost every aspect of the manuscript is in my view inadequately developed or/and presented. The methods -particularly those used for the assessement of the total uncertainty- of the measurement are inacurrate, while the presentation of the results lacks clarity. Additionally, it is troubling to see that based on the results the authors draw conclusions that I would argue are wrong. Lastly, nomeclatures with respect to water isotope analysis and SMOW/SLAP calibrations, some of the presented mathematical expressions as well as nearly every figure in the manuscript need modifications in order to come up to the standard of a journal like AMT. Some of these issues also indicate a possible misunderstanding of some of the concepts or tools presented in the manuscript.

In the following points I will try to explain my decision in more detail to the authors. I would also like to suggest possible approaches and techniques the authors could consider that hopefully can improve the quality of modified version of this manuscript or can be used for compiling a manuscript for a new submission. I would be happy to elaborate more on these ideas in case the authors are interested in using some of those in another version of their work.

**2 General comments**

**2.1 Water isotopic standards**

Accurate measurements of water isotope ratios require properly and accurately calibrated "local" standards with isotopic values on the SMOW/SLAP scale. Quality of

calibrations, storage and handling is essential for the quality of the measurements. I find it very concerning that in this manuscript the values of the two primary standards used for the SMOW/SLAP calibrations are signifficantly different between 2013 and 2014. The differene is more profound for $\delta D$ and up to 5 ‰ difference between the two years. Looking carefully into the reported values in Table 1 it does not look like this large difference is due to evaporation ($D_{xs}$ is roughly unchanged). No uncertainty estimation is given for the discrete measurements either. These differences are concerning and better addressing these issues with the standards is essential and describing them with clarity is necessary.

One very important point here is that there is a claim in the manuscript that the 2014 values are likely more accurate because they represent better the melting conditions of the CFA system since the calibration was carried out in continuous mode (as apposed to the 2013 standard measurements carried out in discrete mode). This is a claim that is fundamentally wrong. As long as an uncertainty estimate is obtained for a measuerement one should not expect to gain anything by calibrating the standards on the same system a measuremnt is performed. This wrongly drawn claim is given in several spots in the manuscript, including the main conclusions (P4line33, P5line3, P8line18, P9line13).

**2.2  Statistically independent errors assumption**

Equation 1 (equation 4 in the manuscript) is essentially a wrong approach in reaching a "total uncertainty" estimate for the system. This is the cornerstone of the uncertainty assessement in this manuscript. It is described as novel and the majority of the reaults and conclusions are based on the application of Eq. 1. Essentially Eq. 1 is the mathematical description of error propagation for the sum of three Gaussian–distributed variables. Equation 1 reads (in my review I will be using the $\sigma$ notation for standard deviation and the $\varepsilon$ notation for the variances as $\varepsilon = \sigma^2$ ; the manuscript mixes those

two symbols and definitions at several places; the authors should make sure they use a consistent nomeclature with respect to variances and standard deviations throughout the manuscript):

$$\sigma_{tot} = \sqrt{\sigma_{Allan}^2 + \sigma_{scatter}^2 + \sigma_{calib}^2} \tag{1}$$

Running the risk of sounding trivial this equation is based on

$$\sigma_Y^2 = \left(\frac{\partial Y}{\partial X_i}\right)^2 \sigma_{x_i}^2 \tag{2}$$

Getting from eq.2 to eq.1 requires an important condition. This is that the variables of which the variances are used in eq. 1 are independent from each other. This can by no means be said for the variables given here. Putting in the same bucket an estimate of the variance of the isotopic composition (in some way what the $\sigma_{scatter}^2$ describes), the error of the mean (essentially what $\sigma_{calib}^2$ represents) and the Allan variance is not an option. All these three parameters are dependent on each other. Therefore I have a hard time seeing how any of the "total uncertainty" estimates in the manuscript is valid. Considering that the manuscript's goal is solely to give proper estimates of the uncertainty I hope the authors can see why I am inclined towards a fresh submission following a very different approach/method.

**2.3 The calculation of $\epsilon_{scatter}$**

This is a questionable calculation in the way it is performed. The quantity $\epsilon_{scatter}$ in the manuscript is essentially a standard deviation and not a variance so here I rewrite the equation using a standard deviation symbol and replacing the "mean" with what it means mathematically. As a result of this eq 8 in the manuscript becomes:

$$\sigma_{scatter} = \frac{1}{N} \sum_1^N \frac{\sigma_i}{\sqrt{n_i}} \tag{3}$$

In eq. 3 $\sigma_i$ is the standard deviation of every 15 sec interval. Firstly the 15 sec choice for the length of the interval is fairly arbitrary and in that sense it introduces a subjectivity in the calculation of $\sigma_{scatter}$. A very important question here that the manuscript does not touch upon at all is what is the influence of the data acquisition rate of the instruments. Calculating standard deviations over 15 s intervals for two instruments that have a different acquisition rate implies that there is a different amount of averaging in the calculation. Is then the difference in $\sigma_{scatter}$ due to actual lower measurement noise of the laser instrument or is it an artifact of the higher acquisition rate (it can also be a combination of these two parameters)? A technical paper like this should deal with these questions when its sole purpose is to characterise the uncertainty of the measurement.

The plots of the scatter error in Figure 5 look very unphysical to me. I suspect that this is due to the fact that you have calulated the scatter error based on the RICE standard blocks bracketing each run and assumed that this error should vary linearly for every data point in between. Even if your system does not have a stricktly Gaussian behavior, to assume that the error varies linearly from one RICE block to the next is by all means statistically wrong.

2.4   The interpretation of the Allan variance

The Allan variance is a great tool for accessing the stability of anaytical instrumentation. Sometimes unfortunately its meaning can be misunderstood. In the case of this manuscript it is misued. Firstly, Fig. 4 does not give any chance to the reader to get an insight on the stability performance of the system based on the Allan variance. It is a Figure with several plots all on grayscale with no legents, or caption explanations as to what is what. What am I looking at? Are these Allan variances of $\delta$D, $\delta^{18}$O  or $\delta^{17}$O? Which plot refers to the 2013 and which one to the 2014 system? The bottom black line shows an Allan variance that decreases almost indefinately. How do you fit that

one when there is no minimum? Why does this curve show such a stable behavior and what is the reason for it? In fact this is a grayscale plot directly taken from Emanuelsson et al. (2015) refering to $\delta^{18}O$ and in fact the bottom curve is from a completely different system that uses a different analyser and sample evaporation system from a different laboratory.

The interpretation and use of the Allan variance in order to calculate the error introduced due to instrumental drifts is falsely performed. The Allan variance gives an estimate of the maximum time one can decrease the error of the mean by averaging. After the optimal point, further averaging is either not helping or makes things worse. So to start with, in the system described in the manuscript there is no averaging of data for such long times (order of 600 s). I would agree that the Allan variance gives an estimate of how your accuracy may be affected by instrumental drifts when SMOW/SLAP calibrations are performed infrequently. This is though only a qualitative assessement as the Allan variance concerns the averaging of data in order to reduce the error of the mean of a variable.

Based on the way that calculation of the Allan variance fit is described and the text "where t is the time difference between the data point and the calibration" I understand that the starting point of your linear fit (where $t = 0$) is the optimal time where the minimum of the Allan variance is located. The error that takes place here is twofold. Firstly for $t = 0$ based on eq. 7 in the manuscript $\varepsilon_{Allan} = 0$. This is not possible. Secondly by starting your fit at the minimum Allan variance you have completely neglected all the first part where averaging in fact does reduce the error of the mean. With this in mind the parabola shape of the Allan error is wrong. It essentially suggests that the error increases monotonously from the time the calibration is finished until the point right in between the two calibrations, when it starts decreasing again monotonoulsy. One has to expect that all measurements that are taken sooner than the optimal time (Allan variance minimum) in fact benefit from averaging. One additional comment on Fig. 4b is that either because of a typo or a miscalculation the legend is wrong. If the

scatter noise for $\delta^{18}O$ is a factor of $\approx 3$ lower than the $\delta D$ signal, then it is physically very difficult (i would say impossible) for a measurement system that measures those two parameters practically simultaneously to result in an Allan variance that is lower for $\delta D$.

**2.5 The explanation of the calibration protocol**

The explanation is rather poor. The terms calibration and normalisation are used wrongly in the manuscript. The term scale normalisation actually refers to the slope. The term calibration refers to the intercept (the term intercept is actually never mentioned throughout the manuscript). Please give the general equation of a calibration and avoid writing terms like "slope" or "$RICE_{true}$" in a mathematical equation. It makes the reading of math formulas very difficult. "the normalisation correction is the measured mean of the RICE standard" Here you probably mean the calibration correction ie line intercept is the value of the RICE standard. This is actually wrong and surprisingly it is not even supported by eq. 3 the way you write it. For a line calibration $\delta_{cal} = a\delta_{raw} + b$ you can solve for $b$ to get your calibration value.

In eq. 3 you define what $step$ is (use directly $t_1$, $t_2$ directly in the equation or replace $step$ with a symbol) but it is unclear what $t$ exactly means. Yes $t$ is the time of the measurement. What units? Is it sec? Using eq. 3 I equalised the two calibration terms from the two different calibrations and solved for $t$ after substituting $t_1$ and $t_2$ in the equation. So the time at which both calibrations are considered equally in the weighing scheme of eq. 3 (this time should fall precisely in the middle between the two calibrations) is $t = 1/2$. Is this 0.5 sec or?? You are likely confusing rather than helping the reader with this type of errors.

**2.6   The quality of the plots**

Nearly all plots are of poor quality. There is no consistency with respect to line coloring (please choose one color for $\delta$D and one color for $\delta^{18}$O and be consistent with your choice), the two parameters ($\delta$D and $\delta^{18}$O) often share a common axes resulting in the very odd "$\delta$" label, measurement units are sometimes in parenthesis (as they should) and sometimes they are placed right next to the $\delta$ symbol with no spacing, while font sizes vary between the different figures. All the plots where the three different errors are presented (fig 4, 5, 7) use the $\delta$ symbol on their axis when the plot actually presents something else. Only fig. 7 shows the full record while fig. 4 and 5 cover shorter section (and not the same) A plot of the calibration error (accuracy) is notably missing from the manuscript. Presentation quality is key for a manuscript of this type and I consider those issues with the manuscript's figures a major drawback.

**2.7   On instrumental drifts**

I wish to make a short note on the topic of instrumental drifts. Instruments can indeed drift with time sometimes in a nicely linear way that can be corrected for. However often one looks into the combined effect of the laser instrument, the sample preparation system as well as the protocol of the measurement. Sometimes in fact, measures to get a handle on the instrumental drifts can actually "create" those drifts in an artifactual way. The injection of a "check standard" or "drift standard" in frequent intervals can in theory offer insight into the nature of the instrumental drifts and possibly allow for a correction. The danger though is that the very same standards create those drifts as they are injected for a time interval that is too short thus not allowing for a stable value due to memory effects.

The most notable misconception with respect to instrumental drifts is that one calls problems as valve wear or drill fluid contamination or leaks as instrumental drifts. They

are not. If your system is leaky at some point in the line you simply have an unsystematic error for which the Allan variance for example cannot say much about, neither can frequent calibrations be of any help. Some of these claims made in the manuscript are purely speculative. Have you got any evidence that the drill fluid causes a spectroscopic interference in the wavelengths you are measuring with the spectrometer or is this something that you just mention in the absence of any other information or guess on sources of error? There is not even information on the type of drill liquid throughout the manuscript. Claims on those possible sources of error can also be found in the last four lines of the abstract. There is no single sentence, or data plot during the whole manuscript providing any (data or physics based) supporting evidence for these claims.

**2.8  The depth registration**

There is absolutely no comment on the uncertainty of the depth registration. Performing CFA measurements on ice cores is a tedious process. It is in fact mentioned in the manuscript that melting was at times interrupted because of sections of poorer ice quality. It is vital to at least comment on errors on the depth scale if this is to be a manuscript on proper error estimation of this analysis. Information on the melting rates used and even a rough estimate on how they vary throughout the measurement is notably missing from the text.

**2.9  Results, discussion and conclusions section**

Here are some more specific comments regarding the results, discussion and conclusions sections.

- It is mentioned that the calibration error is greatly reduced in 2014. No discussion can be found regarding the reasons as to why this is the case. A discussion on

the technical differences between the 2013 and 2014 systems is notably missing. Are the two instruments operating at the same wavelength region? Is there better control of cavity pressure and temperature? Are there changes in the sample transport and evaporation of the isotope line in the CFA system?

- The mean Allan error cannot be higher for $\delta^{18}$O. If the variance of $\delta$D is higher then the Allan variance follows similarly.

- I have a hard time to see how the scatter error is higher for the 2014 system. Based on the Allan variance plots in Emanuelsson et al. (2015) the noise in the data is lower for the 2014 system for both $\delta$D and $\delta^{18}$O. Why do you not plot the scatter error for the full record?

- The calibration error is of course larger than the scatter error. In the ideal case of a Gaussian distribution and assuming that your scatter error represents the standard deviation of the distribution anything within 2 even $3\sigma$ should be considered as acceptable. If your scatter error is say $0.1$ then an accuracy of $0.2$ even $0.3$ can be expected. This is another indication for arguing that adding the errors the way you do with eq. 4 is problematic.

- It makes very little sense to say that calibrating the water standards with the CFA system would result in a better calibration result because you simulate closer the operating conditions of the measurement. A well calibrated standard is independent of the measurement system. It is the quality of your system, your calibration protocol as well as the storage and handling of the standards that will allow for a good result. This statement also appears in the conclusions and can be very misleading to the reader.

- "...so one explanation for the relatively large error is that our drift correction is poorly adapted to this upper range. Ideally, we would use a quality-control standard that falls between the values of our two calibration standards, RICE and

ITASE". All the drift correction does is weigh the influence of two neighbouring calibrations thus if you have a relatively large calibration error then it is your calibration protocol that has an issue. In this sentence you are also speculating again. What makes you think that your drift correction is poorly adapted to the upper range. Show some evidence. What I am wondering about when I am reading this is that you do have such a standard. If you use WP1 and ITASE for your slope/interscept the RICE will be the middle check standard also falling very close to the measurements' level. So the data is there. Adjust you calibration scheme to use WP1 and ITASE then use RICE as check.

- "The overall system performance became more variable in 2014". Looking in fig 7 it is clear that this statement is false. Your total error has fallen considerably for 2014.

- There is no single piece of evidence presented in the manuscript supporting that ice breaks, drill fluid contamination and leaks or valve degradation are reasons for poor quality data. The drill fluid would likely cause considerable effects in the form of outliers if there was a spectroscopic intereference. Have you looked into the instruments spectra and the quality of the spectral fits (fit residuals are usually saved in laser spectrometers)? All of your error metrics (Allan, scatter and calibration error) are based on some assesement of the RICE and WS1 standards measurement or mq water. Thus how do breaks in the ice affect you total error? Leaks and poor pressure control can be checked by looking into the pressure log of the instrument. Do you have any indication based on these data?

**3  Proposed improvements**

Here I outline some proposed improvements and changes in methods used that the authors are welcome to consider in a new version of the manuscript.
**3.1 Spectral estimate of the noise level**

You can get a precision/noise estimate **directly from your data**. Calculate the power spectral density of a section of your data (say 10 m long). It should slowly decay reaching some sort of plateau for the very high frequencies. Take the level of this "tail" and integrade over the full range of frequencies (the surface under a straight line from $-f_{Nyquist}$ to $+f_{Nyquist}$ will be good enough.) The square root of your integral gives you a direct estimate of the noise level from you data. Do this for the full record. Be aware about the type of spectral density you calculated (single/double sided)

**3.2 Measurement accuracy on WS1**

The values you ontain for your accuracy based on the WS1 measurements should be presented for all your calibrations. Make a plot with all of them.

**3.3 The calibration stability**

Give statistics and plots on the values of the calibration line slope and intercept. This is a very interesting statistic and it is great you have all these data. Consider switching to a scheme where WS1 and ITASE are you calibration standards and RICE is your check. Compare to the current scheme where WS1 is your check.

**3.4 A section on instrument/system differences**

Consider a well written section on the instrumental differences between 2013 and 2014. You have a plethora of data on the instruments. Look into spectral fit quality, cavity pressure and temperature for indications of issues with these parameters. Explain better the differences between the two instruments and make sure you understand

well the impact of the higher acquisition rate of the 2014 instrument.

**References**

B. D. Emanuelsson, W. T. Baisden, N. A. N. Bertler, E. D. Keller, and V. Gkinis. High-resolution continuous-flow analysis setup for water isotopic measurement from ice cores using laser spectroscopy. *Atmos. Meas. Tech.*, 8(7):2869–2883, July 2015.

---

## Referee Comment (RC2) · Anonymous Referee #1 · 13 Mar 2018

Please be aware of a typo in the text of my review Equation

$$\sigma_Y^2 = \left(\frac{\partial Y}{\partial X_i}\right)^2 \sigma_{x_i}^2 \tag{1}$$

should instead be

$$\sigma_Y^2 = \sum_i \left(\frac{\partial Y}{\partial X_i}\right)^2 \sigma_{x_i}^2 \tag{2}$$

---

## Referee Comment (RC3) · Anonymous Referee #2 · 5 Apr 2018

This paper describes in some detail a method of calibrating, and uncertainty estimation, of stable water isotopes in ice cores measured by a continuous flow technique. It is described as a novel approach, but I am not sure that this claim is substantiated given that measurement of gases, chemistry and water isotopes in ice cores is now rather common, and all use somewhat similar methods of calibration, drift correction and error analysis. Perhaps it is true that the uncertainty calculation for water isotope has not been published in such a detailed manner before, and this is where the novelty lies. On the whole, the paper is well written: it reads clearly, and is logically set out, and it is interesting. I am aware that an earlier reviewer has made substantial comments about each of the error estimates, and the method of combining the errors for an overall error estimate. While I largely agree with the details of this earlier review of the manuscript,

[Figure]

I do not find it negates the value of the paper to this journal – the discussion of how to characterise the differing contribution to the error of the measurement will be helpful to many working in the field of continuous measurements who have not yet thought through the error contributions quite as deeply as the authors here. Having worked with water isotope instruments for many years, I cannot help feeling that the total mean error estimate recorded in the abstract is qualitatively about as expected, and refining the method of obtaining it will likely not change the figures substantially. And, for example, a small change to the total error in delD of 0.74 per mille would not substantially alter the (qualitative) interpretation of the sample data, which shows a range of about 70 per mille in figure 2, though to be fair small changes in the individual isotope pair errors has a larger impact on the error on the calculated dxs. But for sure, the method by which the total error was obtained is the whole point of the paper, so I am not recommending acceptance without taking into account the criticism of my colleague with the prior review, which I am not about to repeat here in my comments, but offer some further points to address. (One of the downsides of open reviewing is that it is hard to be completely independent.) Specific comments: Given that this is a paper describing the calibration and error estimate, I really want to know how the local standards themselves (four are mentioned on page 4: MilliQ, RICE, WS1, ITASE) were calibrated, and how often. P4/L30 describes the local standards as the accepted value (and sometimes in the manuscript as the 'true values' – when used as a check standard). Several points here. One: how were these accepted values obtained? Details of the calibration of the local standards against the primary Vienna standards (or intermediates) are needed here. Two: they are described as on the VSMOW/SLAP scale at various points in the manuscript, but if I were to be pedantic, I wonder if they were actually calibrated against VSMOW-2 and VSLAP-2 since the original Vienna standards were exhausted about 10 years ago. I wonder whether the local standards here are calibrated against the primary Vienna standards, or through intermediate secondary standards – this level of detail is needed in a paper focussed on calibration. Three: additional errors creep in when producing and maintaining the local isotope standards – each calibration step

removed from the primary Vienna standard introduces a new accuracy and precision error. This error is not included here at all (it isn't there in equation 4 – the calibration error here, as described in section 2.4.3, means an internal error is calibrating the system, rather than the error of the actual standard). As an example, I've just checked my primary Vienna (SMOW-2 and SLAP-2), and they have uncertainties of +/- 0.02 on del18O, and +/- 0.3 on delD. My commercial standards claim uncertainties of +/- 0.2 on del18O, and +/- 1.8 on delD. Four: there is something rather worrying about the 'accepted value' of the local standards in table 1 which is not explained adequately. Why do the standards change from one year to the next? Ideally, once a local standard is prepared I bulk, and calibrated against a primary standard, the local standard is then sealed in aliquots (preferably heat sealed glass ampoules), stored refrigerated, and only opened when used. The paper mentions that the isotope values depend on the water volume in the cavity (section 2.2). This is handled by removing suspect data where the water volume has drifted away from 20,000 ppm, using a short-term averaging method, and whenever the volume fell below 15,000 ppm. Just to be clear: does this remove any need to calibrate the isotope value with the water volume; is the cutting of suspect data sufficient? Figure 3 and P8, L15-19. There is something really rather odd here. The manuscript (P8, L15-19) claims that the drift correction might be 'poorly adapted to the upper range', and that 'Ideally, we would use a quality-control standard that falls between the two calibration standards, RICE and ITASE.' Yet, there are four local standards available: RICE and ITASE, plus MilliQ and WS1. I see no reason why the 'calibration standards' could not have been WS1 and ITASE (or even MilliQ and ITASE) which would have encompassed the whole range of the samples (figure 3 shows that samples fall above the RICE standard for example), and given one (or two) check standards that are within standard range. Figure 3. Could the legend box be moved off the actual data? Currently, figure 3 amply demonstrates the system drift – all standards are drifting to lighter isotope ratio. But the legend box obliterates the early ITASE standard. Figure 4. The upper panel is poorly produced, with no explanation about the different grey curves, or why one Allan plot does not curve upwards. The

dashed 'discrete precision' line is grey in the legend, and black in the figure. The lower panel shows the Allan error parabola for each stack – it would clarify what is demonstrated here to explain that the lowest part of the dip is the start/end of the stack of ice, and the point at which the calibration is carried out. Abstract and P8, L25-30, Summary. Have you any evidence that drill fluid is influencing the error? What fluid is being used? What is its vapour pressure? Could it actually build up in the instrument, or would it simply evaporate at the low cavity pressure and high temperature and be swept out with the waste gas? Are system leaks and valve degradation really affecting the data? What evidence is there, or is this speculation? (I appreciate the text does say 'could affect....'). Following this last point, figures 5 and 7 don't help me understand the scatter and total error over the full record. The scatter error is greater in 2014, due to poor core quality, build up of drill fluid and system leaks – yes? But the overall error is better in 2014 (figure 7). What is surely need here to help clarify this point is a scatter error figure that encompasses the whole core, rather than the short 2013 section in figure 5. Perhaps best of all would be a single figure that for the whole core demonstrates the total error and the three component errors – is this possible?

---

## Author Response (AR1)

**Authors' response to referee comments on "Calculating uncertainty for the RICE ice core continuous flow analysis water isotope record" by Elizabeth D. Keller et al.**

Referee comments are in *black italic font*. Author response is in blue. Changes made to the manuscript text are quoted below each author response. In addition, all revised figures are copied at the end of this document.

We thank the reviewers for their extensive and detailed comments. We note that Reviewer #1 offered broad support for the concept of the manuscript, but also provided comments that were highly critical. Reviewer #2 offered some useful insights but also noted that their perspective had been coloured by the intensely critical remarks of Reviewer #1. We have produced an extensive response to the review comments as a result. Most notably, we considered many of Reviewer #1's criticisms to be either unclear or unconstructive. We believe that our responses to the review, including the improved clarity of the purpose of the manuscript, have improved the publication and we are grateful for the opportunity provided by the reviewers to do so. Specific responses to all comments follow.

Before responding to the specific comments in the reviews, we note broadly that we have attempted to modify the abstract, introduction and conclusions to clarify that the intent and value of the manuscript is to document our system for calculating uncertainties in CFA campaigns involving multiple complex measurement systems. In such campaigns, it is typically not possible to troubleshoot instruments to maintain ideal performance or attribute degraded performance to specific causes. To reflect this, the end of the abstract has been modified to clarify the purpose the paper, and reduce the apparent emphasis on the possible causes of poor instrument performance.

Uncertainties vary through the dataset and were exacerbated by a range of factors, which typically could not be isolated due to the requirements of the multi-instrument CFA campaign. These factors likely occurred in combination and included ice quality, ice breaks, upstream equipment failure, contamination with drill fluid, and leaks or valve degradation. We demonstrate our methodology for documenting uncertainty was effective across periods of uneven system performance and delivered a significant achievement in precision of high-resolution CFA water isotope measurement.

A sentence in the introduction has been edited:

However, the simultaneous operation of approximately 7 measurement systems (Winstrup et al., in review; Pyne et al., 2018) and continuous nature of CFA poses challenges for calibration and uncertainty estimation.

The first paragraph of the conclusion now ends with:

This represents a significant achievement in precision of high-resolution CFA water isotope measurement, and documentation of uncertainty calculations for isotope analyses in a continuous measurement campaign comprising multiple complex measurement systems.

We believe that our responses to review, including the improved clarity of the purpose of the manuscript, bring the manuscript into a form acceptable for publication. Specific responses to all comments follow.

Authors' response to RC #1

*1 Overview*

*The manuscript by Keller et al. presents methods to assess the uncertainty of the water isotope ratio measurement for an ice core continuous flow analysis (CFA) system used for the measurements of the RICE ice core. This is a work that builds on previously published methods by Emanuelsson et al. (2015). The work focuses on a rather special but very essential part of CFA system for water isotopic analysis that of the uncertainty characterisation. It fits very well within the scope of the Atmospheric Measurements Techniques journal. I find it very positive that the authors decide to focus a separate manuscript for assessing the uncertainty of the water isotopic measurement. This is not a usual practice and it is most welcome.*

We thank the reviewer for these overarching comments and support for the value of a separate manuscript assessing uncertainty as a good contribution to *Atmospheric Measurement Techniques.*

*Unfortunately though i cannot recommend the manuscript for publication. I would like to let the decision to the editor (based also on the assessement of the other reviewers) on weather the authors should proceed with a major revision of the manuscript or withdraw their submission and start fresh. The second option would actually be my recomendation. The reason for this decision is that almost every aspect of the manuscript is in my view inadequately developed or/and presented. The methods -particularly those used for the assessement of the total uncertainty- of the measurement are inacurrate, while the presentation of the results lacks clarity. Additionally, it is troubling to see that based on the results the authors draw conclusions that I would argue are wrong. Lastly, nomeclatures with respect to water isotope analysis and SMOW/SLAP calibrations, some of the presented mathematical expressions as well as nearly every figure in the manuscript need modifications in order to come up to the standard of a journal like AMT. Some of these issues also indicate a possible misunderstanding of some of the concepts or tools presented in the manuscript.*

*In the following points I will try to explain my decision in more detail to the authors. I would also like to suggest possible approaches and techniques the authors could consider that hopefully can improve the quality of modified version of this manuscript or can be used for*

*compiling a manuscript for a new submission. I would be happy to elaborate more on these ideas in case the authors are interested in using some of those in another version of their work.*

*2 General comments*

*2.1 Water isotopic standards*

*Accurate measurements of water isotope ratios require properly and accurately calibrated "local" standards with isotopic values on the SMOW/SLAP scale. Quality of calibrations, storage and handling is essential for the quality of the measurements. I find it very concerning that in this manuscript the values of the two primary standards used for the SMOW/SLAP calibrations are signiffically different between 2013 and 2014. The differene is more profound for δD and up to 5 ‰ difference between the two years. Looking carefully into the reported values in Table 1 it does not look like this large difference is due to evaporation (Dxs is roughly unchanged). No uncertainty estimation is given for the discrete measurements either. These differences are concerning and better addressing these issues with the standards is essential and describing them with clarity is necessary.*

The uncertainties for discrete values have been added to Table 1. We apologize for this oversight. The differences between the 2013 and 2014 measurements of WS1 are within the uncertainty range. The differences in RICE and ITASE standard values from 2013 to 2014 are larger. However, it is important to emphasise that these are local laboratory working standards. In our view, the reviewer has misused the term "primary standards", which generally refers to the international primary reference standards, such as VSMOW and VSLAP. The international primary standards are expected to have stable, consistent values over time and in different laboratories.

Local working standards, in contrast, are selected to be of an appropriate material and range of isotope ratios comparable to the samples being analysed. Hence, we chose RICE and ITASE snow as calibration standards encompassing the majority of the sample range and WS1 as a quality control check. Stability is only required within the aliquot, or within batches if storage of identical aliquots from a calibrated batch was undertaken properly. Each batch is carefully calibrated to primary international standards. (In our laboratory, this calibration occurs through a separate set of secondary standards that are stored carefully and calibrated to the VSMOW-2/SLAP-2 scale. These are CM1, SM1 and INS-11, and have

values as described in inserted text copied below. Please see our response to reviewer #2 for further discussion).

It is common (and even expected) during long measurement campaigns such as this one that working standards will change, even when properly prepared and calibrated, as it is not practical to prepare or store all of the necessary material in one batch. We operated our campaign with the assumption that standards can and do change between batches and aliquots. This is one reason that we measured and calibrated our standards to the VSMOW-2/SLAP-2 scale in each individual year, assigning a separate "true" value for 2013 and 2014 in open acknowledgement that there could be considerable differences.

We have added some text explaining our approach:

Accepted or "true" values for these standards as measured against the VSMOW-2/SLAP-2 scale are in Table 1. Each batch of working standards was calibrated to the International Atomic Energy Agency (IAEA) primary standards VSMOW-2 ($\delta^{18}$O = 0.0 ‰; $\delta^{2}$H = 0.0 ‰), SLAP-2 ($\delta^{18}$O = -55.50 ‰; $\delta^{2}$H = -427.5 ‰), and GISP ($\delta^{18}$O = -24.76 ‰; $\delta^{2}$H = 189.5 ‰) using three intermediate, secondary standards INS11 ($\delta^{18}$O = -0.37 ‰; $\delta^{2}$H = -4.2 ‰), CM1 ($\delta^{18}$O = -16.91 ‰; $\delta^{2}$H = 129.51 ‰) and SM1 ($\delta^{18}$O = -28.79 ‰; $\delta^{2}$H = -225.4 ‰).

We note that there is a difference in the true values for RICE and ITASE between 2013 and 2014. We emphasize here that our standards are local working standards, selected or mixed to match the isotope ratios of the sample (melt stream). It is not unexpected that their isotopic value will change between batches during or between long measurement campaigns, as it is not practical to prepare and store all of the material in one batch.

*One very important point here is that there is a claim in the manuscript that the 2014 values are likely more accurate because they represent better the melting conditions of the CFA system since the calibration was carried out in continuous mode (as apposed to the 2013 standard measurements carried out in discrete mode). This is a claim that is fundamentally wrong. As long as an uncertainty estimate is obtained for a measuerement one should not expect to gain anything by calibrating the standards on the same system a measuremnt is performed. This wrongly drawn claim is given in several spots in the manuscript, including the main conclusions (P4line33, P5line3, P8line18, P9line13).*

The reviewer's very strong statement, "fundamentally wrong," is puzzling and appears to either indicate a lack of understanding of what is considered good laboratory practice for isotope standardisation, or a misinterpretation of what we have done and explained. It is very useful here to review the former. An underlying reason for developing this manuscript is to clarify principles of good laboratory practice for continuous isotope measurements using laser spectrometry. Historically, good laboratory practice was passed down in the laboratory

itself; with the advent first of continuous-flow stable isotope mass-spectrometry systems, and now laser spectrometry systems, stable isotope science is accessible to most research groups, and it becomes essential to describe good practice for stable isotope standardisation in the context of the long-standing principles.

We refer the reviewer to the "identical treatment" (IT) principle described in (Werner and Brand, 2001). This describes the importance of identical treatment of both sample and reference material in obtaining accurate results in stable isotope analysis. This includes sample preparation, measurement technique and data processing. The IT principle is recognized as a fundamental part of good laboratory practice for stable isotope analysis (Carter and Fry, 2013; Meier-Augenstein, 2017), as there are numerous factors that could introduce differences in the measured absolute value when any part of the system differs. Only relative measurements of sample and reference material can be considered reliable.

We return to the point made above that our references are local in-house working standards. Previous studies have highlighted that laboratory conditions, sample preparation and measurement pathways can affect the δ value obtained (Brand et al., 2009; Wassenaar et al., 2008). This is true of the measurements made to calibrate the working standards themselves. The more consistent the treatment of all sample and reference material, the more reliable the results will be. We maintain that we can be more confident in the calibration of our working standards in 2014 because we calibrated them to the VSMOW-2/SLAP-2 scale using the same continuous flow analysis system that they were measured with during a calibration cycle in the ice core measurement campaign.

We have clarified the text describing our calibration:

Part of the difference in values might be attributed to the difference in measurement systems. The accepted values for the 2013 calibrations were determined using discrete laser absorption spectroscopy measurements on an Isotope Water Analyzer (IWA) 35EP system. In 2014, our instrument was upgraded with a second laser to IWA-45EP, and the 2014 calibrations utilize values from standards measured continuously with this system. We were regrettably not able to calibrate our working standards using the 2013 CFA setup after the setup was modified for the 2014 campaign, so we use the 2013 discrete measurements in the 2013 calibrations. We thus consider the 2014 melting campaign to be better calibrated than the 2013 campaign. This follows from the principle of "identical treatment" (IT) of stable isotope analysis wherein samples and reference materials should be subject to identical preparation, measurement pathways and data processing to the extent possible (Werner and Brand 2001; Carter and Fry 2013; Meier-Augenstein 2017).

*2.2 Statistically independent errors assumption*

*Equation 1 (equation 4 in the manuscript) is essentially a wrong approach in reaching a "total uncertainty" estimate for the system. This is the cornerstone of the uncertainty assessement in this manuscript. It is described as novel and the majority of the reaults and conclusions are based on the application of Eq. 1. Essentially Eq. 1 is the mathematical description of error propagation for the sum of three Gaussian–distributed variables. Equation 1 reads (in my review I will be using the σ notation for standard deviation and the ε notation for the variances as $\epsilon = \sigma^2$; the manuscript mixes those two symbols and definitions at several places; the authors should make sure they use a consistent nomeclature with respect to variances and standard deviations throughout the manuscript):*

$$\sigma_{tot} = \sqrt{\sigma_{Allan}^2 + \sigma_{scatter}^2 + \sigma_{calib}^2} \qquad (1)$$

We have changed the symbology of the standard deviation and variances as the reviewer has suggested; σ now refers to standard deviations, and variances are denoted $\sigma^2$. This replaces the ε notation, in acknowledgement that this symbol has special significance in isotopes (related to the fractionation factor) and could cause confusion in this context. Additionally, we use notation to distinguish between a value based on the full population of samples (σ) and a value from a sample of the population (*s*) (Kirchner 2001).

We also briefly note that we refer to the web resources cited as Kirchner (2001) for multiple purposes in this response as an openly available and accessible resource, which also underpins the design of our error propagation explained in response to the next comment.

*Running the risk of sounding trivial this equation is based on*

$$\sigma_Y^2 = \sum_i \left(\frac{\partial Y}{\partial X_i}\right)^2 \sigma_{X_i}^2 \qquad (2)$$

*Getting from eq.2 to eq.1 requires an important condition. This is that the variables of which the variances are used in eq. 1 are independent from each other. This can by no means be said for the variables given here. Putting in the same bucket an estimate of the variance of the isotopic composition (in some way what the $\sigma_{scatter}^2$ describes), the error of the mean (essentially what $\sigma_{calib}^2$ represents) and the Allan variance is not an option. All these three parameters are dependent on each other. Therefore I have a hard time seeing how any of the "total uncertainty" estimates in the manuscript is valid. Considering that the manuscript's goal is solely to give proper estimates of the uncertainty I hope the authors can see why I am inclined towards a fresh submission following a very different approach/method.*

We disagree. If we have understood correctly, the reviewer is claiming that there are non-negligible covariances between the different sources of error that we have identified, but has not specified how they are related. This makes it difficult to address their concerns. Critically, we begin by noting that our entire scheme separated components of error which are statistically and analytically uncorrelated in our experimental system. We will outline in some detail the principles behind our method to convince the reader that our error factors are sufficiently independent of one another to a first order approximation.

Firstly, part of the confusion might have arisen from our admittedly casual usage of terms such as accuracy, precision, error and uncertainty. We have endeavoured in the revised text and in our response below to pay careful attention to terminology and its formal meaning in the context of measurement uncertainty (following the conventions described in Analytical Methods Committee, 2003 and Kirchner, 2001). The definitions that we will employ are quoted below from Analytical Methods Committee 2003:

*Accuracy: The closeness of agreement between a test result and the accepted reference value.*

*Trueness: Closeness of agreement between the average value obtained from a large series of test results and an accepted reference value.*

*Precision: The closeness of agreement between independent test results obtained under stipulated conditions … Precision depends only on the distribution of random errors and does not relate to the true value or the specified value.*

Our three error factors can be categorized as follows:

1. "Scatter error": error of the variance / precision / random variation of replicate measurements
2. "Calibration error": error of the mean / trueness
3. Allan error: systematic error due to our imperfect ability to correct for drift

The first two can be quantified from the general analytical techniques given in Kirchner 2001. Note that systematic error or bias does not have a general analytical form; isotopic drift is fortunately amenable to correction, but the method is imperfect.

Furthermore, the scatter error (#1) and the calibration error (#2) are distinguished as follows:

1. Variability in individual sample measurements (SD)

2. Uncertainty in a central mean estimate (SE) of the population or a large set of measurements

The "scatter error", or error of the variance, is a measure of precision. This is the variability in individual sample measurement, sometimes referred to as "repeatability" (how consistent the value is from one point to another when measuring the same sample, without regard to the absolute value). It is usually quantified using the standard deviation of a series of measurements, as we have done (the series of measurements being over a 15 s interval).

Our "calibration error", or error of the mean, is the "trueness" of the average isotopic value when compared to an accepted standard value (rather than the "accuracy" as defined above). The trueness is calculated from the mean of a large set of measurements. We have followed the standard analytical formula for calculating the error of the mean using all of the WS1 calibrations carried out over the measurement campaign. While it is not possible to assume that precision is independent of accuracy in general, it is possible and is accepted practice to separate precision and trueness (Analytical Methods Committee, 2003). Analytically, these are distinct quantities in an estimate of total measurement uncertainty (Kirchner, 2001). The short-term standard deviation (or random variation) of the measurement from one calibration cycle is unlikely in this context to be highly correlated with the error of the mean of the whole population of measurements.

The "Allan error" is a type of systematic error in the system due to the instability of the instrument and its drift over time. We are able to correct for drift to a certain degree through the calibration process, albeit imperfectly. The Allan error is an estimate of how quickly drift changes and therefore how well we are able to correct for it, as a function of time from the nearest calibration. It is derived from the Allan variance, which is commonly used in laser spectroscopy as a tool to quantify instrumental drift (further discussion of Allan variance appears in response to 2.4 below).

In our framework, the systematic bias characterised by the Allan error contrasts with both the scatter and the calibration error. The scatter or noise in the measurements is quantified as random variation over a 15-s interval; instrumental drift is significant on longer time scales. 15 s is typically too short for drift to be noticeable. The calibration error is interpreted as a random component of error in the total uncertainty and is evaluated on the full population of measurements rather than drift over a specific time scale.

We acknowledge that it is impossible for all error factors to be completely independent from one another, as some underlying sources of error will affect all aspects of the system. We

have followed commonly accepted analytical principles and guidelines in identifying and separating different sources of error. We calculated the actual correlation coefficient between the scatter and Allan error for all data points for both isotopes in each year and found them to be uncorrelated ($R^2 < 0.05$). (Note that because the calibration error is a constant it does not make sense to calculate correlation for individual data points).

When uncertainties are correlated, the method of moments may be used to estimate total uncertainty (Kirchner, 2001), as the reviewer has noted. Assuming a moderate estimate of correlation (R = 0.2) and given the mean values for scatter and Allan error, one would add roughly 0.015 ‰ ($\delta^2H$) and 0.005 ‰ ($\delta^{18}O$) to the sum inside the radical. This would add 0.01 ‰ to both of the total error estimates. We argue that we do not see this degree of correlation in our data, and these terms are negligible.

We have clarified the explanation and interpretation of these errors in section 2.4 of the text:

We identified three main sources of uncertainty in our measurements: (i) the Allan variance error (a measure of our ability to correct for drift, a systematic source of uncertainty due to instrumental instability), (ii) the scatter or noise in the data over our chosen averaging interval, and (iii) a general calibration error relating to the overall accuracy of our calibration. Our three error factors can be formally categorized as follows:

1. "Scatter error": error of the variance / precision / random variation of replicate measurements
2. "Calibration error": error of the mean / trueness
3. "Allan variance error": systematic error or bias due to our imperfect ability to correct for drift

The first two can be quantified with general analytical expressions (Kirchner 2001). Systematic error or bias does not have a general analytical form; isotopic drift is fortunately amenable to correction, but the method is imperfect.

We assume that the three error factors are uncorrelated to a large degree. This is supported by the general framework that we have used (Kirchner 2001; Analytical Methods Committee, 2003 ) and the actual errors calculated at each data point ($R^2 < 0.05$ in each year for both isotopes). In practice it is impossible for all error factors to be completely uncorrelated, as some underlying sources of error will affect all aspects of the system. However, we believe these interactions to be small and/or short-lived and negligible to the total uncertainty. With this assumption, we calculate each error factor separately and add them in quadrature to arrive at the total uncertainty estimate:

$$\sigma_{total} = \sqrt{\sigma_{AVE}^2 + \sigma_{scatter}^2 + \sigma_{calib}^2} \tag{5}$$

Each data point in the final record is assigned a unique error value. A detailed explanation of the calculation of each source of uncertainty follows.

*2.3 The calculation of $\varepsilon_{scatter}$*

*This is a questionable calculation in the way it is performed. The quantity scatter in the manuscript is essentially a standard deviation and not a variance so here I rewrite the equation using a standard deviation symbol and replacing the "mean" with what it means mathematically. As a result of this eq 8 in the manuscript becomes:*

$$\sigma_{scatter} = \frac{1}{N} \sum_{1}^{N} \frac{\sigma_i}{\sqrt{n_i}}$$

(3)

Thank you for providing this formula. Eq. 8 in the manuscript has been added to eq. 3 above. However, we have also kept the original formulation. We believe it has value in that it is closer to "machine" language and is therefore easier to translate and implement with computer code. The reviewer has criticized our choice of symbology and formulation of equations in several instances, stating that we have not used the "correct" mathematical symbol and/or that it is difficult to read. We chose in some places to use plain words in lieu of abstract mathematical symbols. We argue that this makes our manuscript more accessible to those without advanced mathematical training and assists in practical understanding and application of the methodology.

*In eq. 3 $\sigma_i$ is the standard deviation of every 15 sec interval. Firstly the 15 sec choice for the length of the interval is fairly arbitrary and in that sense it introduces a subjectivity in the calculation of $\sigma_{scatter}$.*

From a purely mathematical point of view, the length of the interval is arbitrary. This interval was chosen by the RICE project team, from the perspective of climatic interpretation, as the preferred smoothing or averaging interval to apply when analysing the stable isotope record at high resolution. We explored other averaging intervals (30 s and 60 s) but concluded that 15 s was the best at reducing short-term noise without obscuring important features of the record. To be consistent, we have calculated the error using the same time interval that we used to smooth the data in other published work (see e.g. Bertler et al., 2018). Because of this deliberate choice, the error calculation specifically applies over a 15 s interval, which is approximately 7-8 mm on the depth scale.

The reason for this choice is now clarified in the text:

A second error derives from the scatter or noise in the signal over our averaging interval (15 s). This averaging interval was chosen by the RICE project team as a suitable scale to smooth over measurement noise without

obscuring important features in the data. This equates to approximately 7-8 mm on the depth scale. Due to this deliberate choice, the error calculation that follows applies over this interval.

*A very important question here that the manuscript does not touch upon at all is what is the influence of the data acquisition rate of the instruments. Calculating standard deviations over 15 s intervals for two instruments that have a different acquisition rate implies that there is a different amount of averaging in the calculation. Is then the difference in $\sigma_{scatter}$ due to actual lower measurement noise of the laser instrument or is it an artifact of the higher acquisition rate (it can also be a combination of these two parameters)? A technical paper like this should deal with these questions when its sole purpose is to characterise the uncertainty of the measurement.*

We again point out that we are characterising uncertainty over our chosen averaging interval, rather than focusing on instrumental details. The internal data acquisition rate of the instrument is the same in both years; only the rate at which data was recorded and aggregated in the output file differs. It is true that the different data recording rates in 2013 and 2014 result in a different number of data points contained in a 15 s interval in each of the two years. In 2013, the rate was twice that of 2014 (one data point written every 0.5 s versus one every 1.0 s). However, the corresponding depth of the sample being melted was only recorded every 1.0 s in both years. This resulted in duplicated depth assignment in 2013. We reconciled this by averaging all data points (in most cases, 2 data points) assigned to the same depth, in effect giving us the same resolution on the depth scale. It is largely irrelevant whether any difference in scatter is due to different aggregation rates because we are quantifying the uncertainty over the averaging interval actually used.

We have added some detail noting the difference:

We note that the number of points that are contained in the interval is different in 2013 and 2014, resulting from the difference in output aggregation (not the instrument's internal data acquisition rate). This could affect the amount of noise in the data. However, we have not attempted to analyse this in detail, as we are only concerned here with quantifying the uncertainty associated with our averaging interval, regardless of the number of data points averaged.

*The plots of the scatter error in Figure 5 look very unphysical to me. I suspect that this is due to the fact that you have caluclated the scatter error based on the RICE standard blocks bracketing each run and assumed that this error should vary linearly for every data point in*

*between. Even if your system does not have a stricktly Gaussian behavior, to assume that the error varies linearly from one RICE block to the next is by all means statistically wrong.*

We find this comment confusing, in that it does not explain why the reviewer thinks this assumption is wrong nor how to fix it. The linear variation across a stack is a result of the time-weighted averaging applied to the two errors calculated from each of the two calibration cycles on either side of a stack. We believe it is reasonable to assume, to the first order, that the error is a linear combination of the two factors calculated from the two calibrations bracketing the measurements. This assumption is rooted in the fact that the noise in a set of measurements from the same sample can in general be modelled as a Gaussian process with a normal distribution of independent random variables. The standard deviation is akin to a random walk / Brownian motion, and the mean-squared displacement is linear with time. Thus when two end points are known (in this case the two calibrations), the path between them is best estimated by a line, even though unpredictable variations occur between the points of measurement.

We have added some text explaining this:

Again, because the RICE standard was measured both before and after each stack, we calculate for both measurements and linearly combine them using a time-weighted average. Note this error is linear with time within a stack but is discontinuous at the points where a stack begins and ends. This linearity is rooted in the fact that the noise in a set of measurements from the same sample can in general be modelled as a Gaussian process, with a normal distribution of independent random variables. The mean-squared displacement is linear with time. Scatter error vs. depth for the whole length of the core is shown in Fig. 5.

*2.4 The interpretation of the Allan variance*

*The Allan variance is a great tool for accessing the stability of anaytical instrumentation. Sometimes unfortunately its meaning can be misunderstood. In the case of this manuscript it is misued. Firstly, Fig. 4 does not give any chance to the reader to get an insight on the stability performance of the system based on the Allan variance. It is a Figure with several plots all on grayscale with no legents, or caption explanations as to what is what. What am I looking at? Are these Allan variances of $\delta D$, $\delta^{18}O$ or $\delta^{17}O$? Which plot refers to the 2013 and which one to the 2014 system? The bottom black line shows an Allan variance that decreases almost indefinately. How do you fit that one when there is no minimum? Why does this curve show such a stable behavior and what is the reason for it? In fact this is a grayscale plot directly taken from Emanuelsson et al. (2015) refering to $\delta^{18}O$ and in fact the*

*bottom curve is from a completely different system that uses a different analyser and sample evaporation system from a different laboratory.*

This figure was meant as a conceptual illustration of the Allan error. It was not intended to give insight into the stability of the system. The reader is referred to Emanuelsson et al., 2015 for a complete and thorough analysis. The particulars of the grayscale curves are not relevant to the concept presented here, which is why they were not labelled individually. It seems that our intention was not clear and that the amount of curves in the plot was distracting. This panel has been removed from the figure. (See new Figure 4 at the end of this document.)

*The interpretation and use of the Allan variance in order to calculate the error introduced due to instrumental drifts is falsely performed. The Allan variance gives an estimate of the maximum time one can decrease the error of the mean by averaging. After the optimal point, further averaging is either not helping or makes things worse. So to start with, in the system described in the manuscript there is no averaging of data for such long times (order of 600 s). I would agree that the Allan variance gives an estimate of how your accuracy may be affected by instrumental drifts when SMOW/SLAP calibrations are performed infrequently. This is though only a qualitative assessement as the Allan variance concerns the averaging of data in order to reduce the error of the mean of a variable.*

*Based on the way that calculation of the Allan variance fit is described and the text "where t is the time difference between the data point and the calibration" I understand that the starting point of your linear fit (where t = 0) is the optimal time where the minimum of the Allan variance is located. The error that takes place here is twofold. Firstly for t = 0 based on eq. 7 in the manuscript $\varepsilon_{Allan} = 0$. This is not possible. Secondly by starting your fit at the minimum Allan variance you have completely neglected all the first part where averaging in fact does reduce the error of the mean. With this in mind the parabola shape of the Allan error is wrong. It essentially suggests that the error increases monotonously from the time the calibration is finished until the point right in between the two calibrations, when it starts decreasing again monotonoulsy. One has to expect that all measurements that are taken sooner than the optimal time (Allan variance minimum) in fact benefit from averaging.*

We thank the reviewer for providing a description of one important use of the Allan variance. We agree that it can be used to find the optimal averaging time. This, however, is only one

application, and other uses are not precluded. The laser spectroscopy community has embraced the Allan variance as a useful tool to estimate the standard deviation across time intervals over which averaging can be applied, and thus by extension provides information about instrumental drift (which the reviewer apparently accepts). Based on its common usage in laser spectroscopy, we see no valid reason why this metric can be used to estimate uncertainty arising due to drift since the last calibration.

We would like to point out that the term that we refer to as the "Allan error", although derived from the Allan variance, is not equivalent to it. We have fit the increasing part of the Allan variance curve to a line to estimate the trend of increasing standard deviation with time. We take this as an estimate of how quickly drift is increasing with time, and thus a measure of how well we are able to correct for drift using our relatively infrequent calibration measurements. We assume that we know the drift perfectly at the point of the calibration, where we have calculated it (which is not strictly true, but is close at our levels of precision). Consequently, at the time immediately after the calibration (t = 0), this uncertainty is zero. As we get farther away in time from the point where we have calculated the drift, we are less able to correct for it because our knowledge of the amount of drift gets worse. As we near the calibration on the other side of the stack, our knowledge of the amount of drift again improves, until we have another anchor point and direct calculation of drift from the following calibration cycle. We emphasize that we are not describing the uncertainty due to time-averaging.

We have tried to clarify these points:

We calculated the Allan deviation of our system using measurements of the Milli Q standard, run continuously for 24-48 hours. We conducted these tests periodically during both measurement campaigns (usually over the weekend when the instruments were otherwise idle; see Emanuelsson et al., 2015 for details). On a log-log plot of the Allan deviation vs. averaging time ($\tau$), there is a minimum at the averaging time where the precision is highest; before this point, at very short averaging times, instrumental noise affects the signal, and after, at longer averaging times, the effects of instrumental drift can be seen. Thus, the Allan deviation provides an estimate of the optimal averaging time, before and after which precision decreases.

The Allan deviation can also provide an indication of the uncertainty due to instrumental drift as a function of the time difference between the measurement and the nearest calibration. For our system to stay under the precision limit of 1.0 ‰ and 0.1 ‰ for $\delta^2H$ and $\delta^{18}O$, respectively (and to permit analysis with deuterium excess, d = $\delta^2H$ - 8 * $\delta^{18}O$), a calibration cycle to correct for drift should occur at least every ~1 hr during ice core measurements (Emanuelsson et al., 2015). However, as noted above, system limitations prevented us from running calibrations as frequently as would have been optimal. We use the Allan deviation here to estimate how quickly instrumental drift is increasing and thus how well we are able to correct for drift using our calibrations.

*One additional comment on Fig. 4b is that either because of a typo or a miscalculation the legend is wrong. If the scatter noise for δ¹⁸O is a factor of 3 lower than the δD signal, then it is physically very difficult (i would say impossible) for a measurement system that measures those two parameters practically simultaneously to result in an Allan variance that is lower for δD.*

The legend on Fig 4. is correct. The Allan error is lower for $\delta^2H$ than for $\delta^{18}O$. We found in previous work (Emanuelsson et al., 2015) that the peak Allan deviation in the middle of a stack (about 1.2 hrs from the nearest calibration) is 0.17 and 0.13 ‰ for $\delta^{18}O$ and $\delta^2H$, respectively. This appears to be a misunderstanding of the nature of the error factors. The amount of noise is distinct from the systematic error due to drift. We do not see why a lower amount of noise in the $\delta^{18}O$ measurements would prevent the Allan error from being higher than that of $\delta^2H$. It is also worth pointing out that this, in fact, supports our claim that these errors are independent of one another.

*2.5 The explanation of the calibration protocol*

*The explanation is rather poor. The terms calibration and normalisation are used wrongly in the manuscript. The term scale normalisation actually refers to the slope. The term calibration refers to the intercept (the term intercept is actually never mentioned throughout the manuscript). Please give the general equation of a calibration and avoid writing terms like "slope" or "RICEtrue" in a mathematical equation. It makes the reading of math formulas very difficult. "the normalisation correction is the measured mean of the RICE standard" Here you probably mean the calibration correction ie line intercept is the value of the RICE standard. This is actually wrong and surprisingly it is not even supported by eq. 3 the way you write it. For a line calibration $\delta_{cal} = a\delta_{raw} + b$ you can solve for b to get your calibration value.*

We have rewritten the explanation of our calibration procedure using the terminology given in Paul et al. 2007. We followed the "two-point linear normalization" method laid out in this reference, which is now routinely used to adjust measured isotope values to a reference scale (e.g. Munksgaard et al., 2018). This method uses linear regression of two measured and true values of laboratory standards to normalize the measured samples with respect to the isotopic reference scale (here defined by our local working standards). The slope is calculated by plotting the measured values on the x-axis and the true values on the y-axis

and then using trigonometric formulas to relate them to the true value of the sample (Eq. 16 in Paul et al. 2007, reproduced below). Equations 16 and 17 in Paul et al. 2007 are the basis of our correction and identical to our formula, with $\delta_{Std2} = \delta_{RICE}$ and $\delta_{Std1} = \delta_{ITASE}$. The linear regression equation takes the usual form $\delta_{corrected} = m\delta_{measured} + b$. By design, $b$ (the y-intercept) is the true $\delta$-value of the RICE standard. (Here we feel that the reviewer has added to the confusion over terminology conventions. The term "calibration" is used loosely throughout the literature and does not refer to the y-intercept as we have implemented it.) We have added the generic linear regression equation to the text as requested. Paul et al. 2007 explains the two-point normalization procedure with a straightforward illustration. We have reproduced Eqs. 16 and 17 and Figure 2 here for the purpose of clarifying our procedure.

[Figure]

**Figure 2.** Illustration of derivation of Eqn. (16) that is the basis for two-point normalization. The slope ($m=\tan\alpha$) of the regression line, $\overline{ADE}$, is represented by the trigonometric relationship between the base and height of two right-angle congruent triangles ABD and ACE.

$$slope(m) = \tan\alpha = \frac{\delta^T_{Spl} - \delta^T_{Std2}}{\delta^M_{Spl} - \delta^M_{Std2}} = \frac{\delta^T_{Std1} - \delta^T_{Std2}}{\delta^M_{Std1} - \delta^M_{Std2}} \qquad (16)$$

$$\delta^T_{Spl} = \frac{\delta^T_{Std1} - \delta^T_{Std2}}{\delta^M_{Std1} - \delta^M_{Std2}} \times \left(\delta^M_{Spl} - \delta^M_{Std2}\right) + \delta^T_{Std2} \qquad (17)$$

Additionally, please see our comment in response to item 2.3 above on the use of words in formulas (such as "slope" and "RICEtrue"). Nevertheless Eq. 2 in the manuscript has been rewritten as copied in the text that follows.

The revised text reads:

We have used a two-point linear normalisation procedure, which is routinely used to adjust measured δ-values to an isotopic reference scale (Paul et al., 2007). The correction takes the form of linear regression: $\delta_{corrected} = m\delta_{measured} + b$, where $m$ is the slope of the line and $b$ is the y-intercept. The measured δ-values of two laboratory standards are regressed against their "true" δ-values. The slope $m$ can be calculated by plotting the measured values on the x-axis and true values on the y-axis and then using trigonometric formulas to relate them to the true value of the sample (Paul et al., 2007). The result is the ratio of the difference between the true RICE and ITASE δ values and the actual difference measured:

$$m_i = \frac{\delta^T_{RICE} - \delta^T_{ITASE}}{\delta_{RICEi} - \delta_{ITASEi}} \qquad (1)$$

where $\delta^T_{RICE}$ and $\delta^T_{ITASE}$ are the accepted standard values and $\delta_{RICEi}$ and $\delta_{ITASEi}$ are the $ith$ measured value of the standards RICE and ITASE, respectively. The correction then takes the following form:

$$\delta_{corrected} = \frac{\delta^T_{RICE} - \delta^T_{ITASE}}{\delta_{RICEi} - \delta_{ITASEi}} * (\delta_{raw} - \delta_{RICEi}) + \delta^T_{RICE} \qquad (3)$$

By design, the y-intercept $b$ is $\delta^T_{RICE}$. We calculated this correction for each stack using the closest set of RICE and ITASE calibration measurements both before and after the stack. We then apply the correction to each data point by weighting the factors calculated from the calibrations before and after the stack by the time difference between the data point and the calibration:

$$\delta_{corrected}(t) = \left[(\delta_{raw} - \delta_{RICE1}) * m_1 + \delta^T_{RICE}\right] * (1 - f) + \left[(\delta_{raw} - \delta_{RICE2}) * m_2 + \delta^T_{RICE}\right] * f \qquad (4)$$

where $\delta_{raw}$ is the uncalibrated raw $\delta^2H$ or $\delta^{18}O$ value of the ice core sample, $\delta_{RICE1}$ and $\delta_{RICE2}$ are the measured values of the RICE standard before and after the stack, respectively, $t$ is the time of $\delta_{raw}$ measurement relative to $t_1$, and $f$ is a dimensionless weighting factor: $f = t/(t_2 - t_1)$, $t_1$= starting time of $\delta_{raw}$ measurement before the stack, and $t_2$= ending time of measurement after the stack. We note that this method assumes that drift is approximately linear over the measurement period. Our calibration procedure was validated by comparison to discrete measurements in Emanuelsson et al. (2015).

*In eq. 3 you define what step is (use directly t1, t2 directly in the equation or replace step with a symbol) but it is unclear what t exactly means. Yes t is the time of the measurement. What units? Is it sec? Using eq. 3 I equalised the two calibration terms from the two different calibrations and solved for t after substituting t1 and t2 in the equation. So the time at which both calibrations are considered equally in the weighing scheme of eq. 3 (this time should fall precisely in the middle between the two calibrations) is t = 1=2. Is this 0.5 sec or?? You are likely confusing rather than helping the reader with this type of errors.*

There is a typo in Eq. 3: $(1 - t) * step$ should have been $(1 - t * step)$. We apologize if this caused confusion. We have modified the way this equation is written to explicitly include a dimensionless weighting factor *f* (the proportion of the total time between calibrations that has elapsed). The correct equation (now Eq. 4) reads:

$$\delta_{corrected}(t) = \left[(\delta_{raw} - \delta_{RICE1}) * m_1 + \delta_{RICE}^T\right] * (1 - f) + \left[(\delta_{raw} - \delta_{RICE2}) * m_2 + \delta_{RICE}^T\right] * f \qquad (4)$$

We believe we have addressed these concerns by fixing the typo and replacing the *t* symbol (associated with time units) with *f,* a dimensionless quantity.

*2.6 The quality of the plots*

*Nearly all plots are of poor quality. There is no consistency with respect to line coloring (please choose one color for δD and one color for δ¹⁸O and be consistent with your choice), the two parameters (δD and δ¹⁸O) often share a common axes resulting in the very odd "" label, measurement units are sometimes in parenthesis (as they should) and sometimes they are placed right next to the symbol with no spacing, while font sizes vary between the different figures.*

We find this criticism puzzling. The colors in all plots where both $\delta^2H$ and $\delta^{18}O$ and shown are, in fact, the same ($\delta^2H$ is green and $\delta^{18}O$ is black). Where only one isotope but different data streams are shown, different colors are necessary to distinguish the lines from one another. The legends and captions explain the colors in all figures. (Note that because we have redone Fig. 7 and changed the colors, we have also changed the colors in the other figures. In the revised figures, blue / cyan is used for $\delta^2H$ and red / magenta is used for $\delta^{18}O$. In the other plots, raw data is in black (both isotopes) and other colors are used to highlight features as explained in the legend and captions.) It is clear from the legends, captions and description in the text which isotope is being presented. The units on the y-axis are straightforward (in per mil), and sharing a common axis allows for comparison of the

magnitude of the error for both isotopes in the same figure. We do not think that this is odd. The spacing / parentheses around the units has been fixed. Despite the reviewer's impression, the figures were all created in Matlab with the same font for all axes labels (Arial size 14) and tick marks (size 12), with the exception of figures with multiple panels, which are scaled according to available space. The numbers and labels are all readable and can be easily adjusted to fit the final layout at the request of the editors.

*All the plots where the three different errors are presented (fig 4, 5, 7) use the symbol on their axis when the plot actually presents something else.*

We have changed the delta symbol to sigma. (We assume this is what the reviewer had in mind since the comment is not more specific.)

*Only fig. 7 shows the full record while fig. 4 and 5 cover shorter section (and not the same) A plot of the calibration error (accuracy) is notably missing from the manuscript.*

We chose to show only a portion of the record so that fine-scale detail was visible. However, we have reconsidered this presentation in light of both reviewers' comments and have modified the figures to show all depths. The calibration error is a constant in each year. We do not think a plot of a constant is useful in this context. We have added supplementary material (copied at the end of this document) showing the variation in calibrations across the campaign, as the reviewer has requested in 3.2.

*Presentation quality is key for a manuscript of this type and I consider those issues with the manuscript's figures a major drawback.*

The majority of the features that the reviewer has criticized are cosmetic details that are easily modified. We view these as minor corrections. We note that these and other strongly worded comments regarding figures do not contain constructive suggestions which we could action. Where the content of the figures could identifiably benefit from adjustment, we have done so. The new figures are copied at the end of this document.

*2.7 On instrumental drifts*

*I wish to make a short note on the topic of instrumental drifts. Instruments can indeed drift with time sometimes in a nicely linear way that can be corrected for. However often one looks into the combined effect of the laser instrument, the sample preparation system as well as the protocol of the measurement. Sometimes in fact, measures to get a handle on the instrumental drifts can actually "create" those drifts in an artifactual way. The injection of a "check standard" or "drift standard" in frequent intervals can in theory offer insight into the nature of the instrumental drifts and possibly allow for a correction. The danger though is that the very same standards create those drifts as they are injected for a time interval that is too short thus not allowing for a stable value due to memory effects.*

*The most notable misconception with respect to instrumental drifts is that one calls problems as valve wear or drill fluid contamination or leaks as instrumental drifts. They are not. If your system is leaky at some point in the line you simply have an unsystematic error for which the Allan variance for example cannot say much about, neither can frequent calibrations be of any help. Some of these claims made in the manuscript are purely speculative. Have you got any evidence that the drill fluid causes a spectroscopic interference in the wavelengths you are measuring with the spectrometer or is this something that you just mention in the absence of any other information or guess on sources of error? There is not even information on the type of drill liquid throughout the manuscript. Claims on those possible sources of error can also be found in the last four lines of the abstract. There is no single sentence, or data plot during the whole manuscript providing any (data or physics based) supporting evidence for these claims.*

The reviewer seems to have misunderstood our explanations of possible sources of error. We have not said that instrumental drift can be attributed specifically to drill fluid, leaks, or anything else. We have not claimed that we can correct for all types of error with calibrations. On the contrary, what we have tried to do is to develop a method of calculating uncertainty that does not require source attribution in every instance.

We reiterate our philosophy that with major instrumentation or operation campaigns of this sort, it will not always be possible separate causes of drift or variance; but it is essential to quantify drift and variance, as we have done here. Our guide for what to include in the manuscript is a description of plausible sources of enhanced drift and variance, without venturing into attribution of these terms to particular issues. From a more practical perspective, many operational systems will have a leak at some point. As an example, a small leak is likely to be inseparable from some level of background requiring correction that

is a given in many systems. A reasonable analyst will try to keep going and correct when this issue is below a certain threshold that would undermine the usability of results. This is particularly valid in a measurement campaign such as ours, where downtime for troubleshooting means that 6 other measurement systems are required to go idle.

We invite the reviewer to reread the last two paragraphs of Section 3 in which we discuss degradation in performance (not drift specifically), particularly in 2014. Our description stems from some of our observations "on the ground" during the measurement campaign. These are offered as examples of things that might have gone wrong (and probably did) but could not be verified in the middle of a major operation. There were significantly more performance issues in 2014, at the point in the core where we encountered brittle ice (> 500 m). We noticed that there was more variability in measurement noise in these deeper layers of the core, where breaks and microfractures were more frequent and drill fluid contamination in the inner part of the core became more likely. Drill fluid was definitely present in the melt water, as contamination was obvious in other CFA measurements such as dust particle counts (see Warming et al., 2013 for documented effects of Estisol on some types of CFA measurements). There was a visible build-up of residue in the nebulizer; after cleaning the chamber and tightening the valves, the system became more stable (i.e. less noise in the measurements) for a short time before once again declining over the course of a few days. We have clearly stated that we "have only anecdotal evidence" with respect to the increase in noise.

For a further discussion and analysis of the influence of drill fluid, please see our response to reviewer #2. We have added the composition of the drill fluid to the text.

It is generally not practical during lengthy ice core measurement campaigns to stop the campaign and identify and separate the sources of instability or poor performance every time. We agree with the reviewer that the reasons behind this can be unsystematic sources of error in our system. We have tried to offer some likely explanations for the observed decline in system performance, but note that these cannot be mapped directly onto our uncertainty calculations. We would like to emphasize that this manuscript is not about the particulars of our measurement campaign but rather a more general methodology to deal with unpredictable sources of error.

The last two paragraphs of Section 3 are now as follows:

The overall system performance became more variable from day to day in 2014, despite the decrease in total error. There are three main possible reasons for the large variations in performance. They are: 1) response to breaks in ice and associated bubbles; 2) performance degradation due to unexpected levels of drill fluid in the melt stream (a mixture of Estisol-240 and Coasol was used to keep the drill hole open; although all pieces of ice

were thoroughly cleaned before melting, some contamination occurred through existing microfractures in the ice); 3) leaks or valve degradation in the laser spectrometer, which operates under vacuum. There were significantly more performance issues in 2014. In addition to the different setup and gradual build-up of drill fluid in the instruments over time, the ice itself was of poorer quality at deeper depths (in the brittle ice zone at depths below 500 m; Pyne et al., 2018), containing more breaks that caused interruptions in the CFA measurements and possible drill fluid contamination. Although we have only anecdotal evidence, the more frequent stopping and restarting of the system in 2014 seemed to introduce more noise into the measurements.

Because the campaign was conducted to operate many measurement systems simultaneously, as is characteristic of ice core CFA campaigns, it was typically not possible to conduct comprehensive performance tests and systematic evaluations during the one day of down time in each week-long, seven-day cycle. As a result, the precise sources of performance deterioration were difficult to isolate. Our method for calculating uncertainty is designed to reflect the changing day-to-day conditions without the need to attribute the exact source of error.

*2.8 The depth registration*

*There is absolutely no comment on the uncertainty of the depth registration. Performing CFA measurements on ice cores is a tedious process. It is in fact mentioned in the manuscript that melting was at times interrupted because of sections of poorer ice quality. It is vital to at least comment on errors on the depth scale if this is to be a manuscript on proper error estimation of this analysis. Information on the melting rates used and even a rough estimate on how they vary throughout the measurement is notably missing from the text.*

Depth alignment across multiple measurement systems is indeed an important issue in an ice core campaign. The age scale is determined through a combination of annual layer counting of the CFA stable isotope dataset (in the top portion of the core) and CFA methane gas measurements (Windstrup et al., in review; Lee et al., in prep), so in order to accurately determine the age chronology, the depth must be carefully aligned across all instruments. This is particularly important deeper in the core, where a misalignment of a few centimeters could equate to hundreds or even thousands of years (Lee et al., in prep). The identification of key events in the climate history depends to some extent on accurate depth alignment. Consequently, a great deal of effort must be devoted to aligning all instrument data to a common depth scale.

Some of the reviewer's concerns have been addressed in Pyne et al., 2018 (which we appreciate was in review at the time we submitted this manuscript). The melting rate was approximately constant at 3 cm min$^{-1}$ over the course of the campaign, producing a liquid flow rate of ~16.8 mL per minute. This information has been added to the text. We have

added a paragraph describing our method of depth alignment and a very rough estimate of the uncertainty. Error was mainly introduced due to the lag between the time the ice was melted (where the depth was recorded) and the time the meltwater reached the IWA, and the imprecise nature of identifying the transition between the calibration cycle and the beginning of an ice core stack in the stable isotope data. Error in the time lag would result in a systematic shift in the isotopic values in a stack. However, we view quantifying this error more precisely to be outside the scope of the manuscript.

Poor quality ice also affected the depth registration, particularly in the brittle ice zone (below 500 m depth) where fractures were more frequent. An optical depth encoder rested on top of the melting ice stack and recorded vertical displacement. The breaks and fractures in the ice occasionally caused a portion of the stack to get stuck; in these situations, the depth encoder failed to register any change in depth for a time (while the lower portion of the stack continued to melt). We linearly interpolated over these intervals, assuming a constant melt rate. This introduced a small amount of uncertainty in the depths assigned to those timestamps, but we believe it was negligible given that the melt rate did not vary much during the campaign.

We estimate, at most, that these factors introduced errors of 1-10 mm in the depth alignment.

We have added some text discussing these issues:

The ice was cut into 1 m segments and melted at a controlled rate of approximately 3 cm min$^{-1}$, producing a liquid flow rate of ~16.8 mL per minute (Pyne et al., 2018). …

Breaks in the ice were measured and recorded to 1.0 mm precision before melting. Any ice that was cut out and removed was recorded as a gap in the depth scale. Processing of the raw data files was performed using a graphical user interface (GUI) and a semi-automated script in Matlab (Matlab Release 2012b, The MathWorks, Inc., Natick, Massachusetts, United States). Occasionally, poor-quality ice (i.e ice containing fractures and slanted breaks) caused the upper part of the stack to stick to the sides of the core holder; the depth encoder failed to register any change in depth for a time, while the base of the stack continued to melt. These intervals required linear interpolation (assuming a constant melt rate) and introduced a small amount of uncertainty (Pyne et al., 2018). This occurred more frequently deeper in the core in the brittle ice zone ( > 500 m). Given that the melt rate was fairly constant throughout the campaign, the error introduced in the depth assignment was negligible. More details of the data processing are available in Pyne et al. 2018. …

The campaigns together required processing and alignment of over 5 million raw data points. Depth alignment across multiple measurement systems is a key issue for ice core campaigns and a fundamental requirement for producing an age chronology (Winstrup et al., in review). The interpretation and identification of key events in the climate history thus depends on accurate depth alignment. This is particularly important deeper in the core,

where a misalignment of a few centimeters could equate to hundreds or even thousands of years (Lee et al., submitted).

Alignment of the isotope data to the depth scale is based on the time lag between the depth log file and the WVIA instrument output. The time lag was determined with an automated algorithm to detect the end of the calibration cycle and the beginning of the ice core melt stream from the abrupt change in numeric derivatives of adjacent data points. The calculated time lags during each measurement campaign averaged 418 s in 2013 and 156 s in 2014, but varied slightly from day to day by 10-20 s. (The lag was shorter in 2014 due to the reduction in length of tubing between the melter and WVIA. Variations occurred from the periodic replacement of the tubing.) There were a few occasions of equipment failure where manual depth alignment was necessary. As mentioned above, poor ice quality also affected the depth logfiles (Pyne et al., 2018). The precise quantification of the uncertainty introduced from the depth assignment is beyond the scope of this manuscript; based on the variation in time lags, we estimate that, at most, it is on the order of 1-10 mm.

Where the added text and our comments above perhaps do not completely address the reviewers concern that the "tedious process" of depth registration has not been fully explained, we reiterate that it is not the primary subject of this manuscript but is a focus of additional manuscripts on this measurement campaign.

*2.9 Results, discussion and conclusions section*

*Here are some more specific comments regarding the results, discussion and conclusions sections.*

*• It is mentioned that the calibration error is greatly reduced in 2014. No discussion can be found regarding the reasons as to why this is the case. A discussion on the technical differences between the 2013 and 2014 systems is notably missing. Are the two instruments operating at the same wavelength region? Is there better control of cavity pressure and temperature? Are there changes in the sample transport and evaporation of the isotope line in the CFA system?*

Most of this information can be found in Emanuelsson et al. (2015). However, for clarity we have added a short description of the main differences. The setups in 2013 and 2014 are largely the same. The instruments did operate at the same wavelength region. In 2014, the vaporizer heating element was modified. There was also a higher sample flow of    150 µL

min−1 matched with dry air flow (to achieve the same    20,000 ppm water vapor

concentrations as in 2013), and the mixed vapour was delivered directly to the IWA via an open split, rather than through the split contained in the WVISS.

*• The mean Allan error cannot be higher for $\delta^{18}O$. If the variance of $\delta D$ is higher then the Allan variance follows similarly.*

The quantity that we have termed the "Allan error" is not equivalent to the Allan variance. Our error term is a measure of how quickly the Allan variance increases with time, not how large that variance actually was. Please see our answer to 2.4 above.

*• I have a hard time to see how the scatter error is higher for the 2014 system. Based on the Allan variance plots in Emanuelsson et al. (2015) the noise in the data is lower for the 2014 system for both $\delta D$ and $\delta^{18}O$. Why do you not plot the scatter error for the full record?*

We have now included all depths in Figure 5. We think this addresses some of the reviewer's concerns. As we state in the manuscript, the instrumental performance in 2014 was highly variable; at times the scatter error in 2014 was lower than in 2013, but on average it was higher. There are several instances in 2014 where performance suddenly deteriorated; this can be seen as a short spike in the scatter error (e.g. depths 505-508 m). The Allan variance plots in Emanuelsson et al. (2015) are not indicative of the variability that we saw in instrument performance across the whole campaign. Again, the scatter error is distinct from the Allan variance error.

*• The calibration error is of course larger than the scatter error. In the ideal case of a Gaussian distribution and assuming that your scatter error represends the standard deviation of the distribution anything within 2 even 3σ should be considered as acceptable. If your scatter error is say 0:1 then an accuracy of 0:2 even 0:3 can be expected. This is another indication for arguing that adding the errors the way you do with eq. 4 is problematic.*

We are not sure why this indicates that adding the error factors is problematic. The scatter error is a short-term standard deviation. The calibration error is the error of the mean of a large set of measurements. These are analytically distinct quantities. It is possible to have a large standard deviation (low precision) but still have high accuracy in the mean and vice

versa. We refer to our explanation in response to 2.2; the reviewer has not explained their reasoning sufficiently for us to provide a more specific answer here.

• *It makes very little sense to say that calibrating the water standards with the CFA system would result in a better calibration result because you simulate closer the operating conditions of the measurement. A well calibrated standard is independent of the measurement system. It is the quality of your system, your calibration protocol as well as the storage and handling of the standards that will allow for a good result. This statement also appears in the conclusions and can be very misleading to the reader.*

We have defended our claim in response to 2.1 above. It is true that system quality, calibration protocol and sample handling all can affect the accuracy of the results. However, the principle of identical treatment is fundamental for stable isotope analysis. A calibrated standard is not independent of the measurement system, as numerous laboratory intercomparison studies have shown. To suggest otherwise would in fact be misleading and contrary to good laboratory practice.

• *"...so one explanation for the relatively large error is that our drift correction is poorly adapted to this upper range. Ideally, we would use a quality-control standard that falls between the values of our two calibration standards, RICE and ITASE".* All the drift correction does is weigh the influence of two neighbouring calibrations thus if you have a relatively large calibration error then it is your calibration protocol that has an issue. In this sentence you are also speculating again. What makes you think that your drift correction is poorly adapted to the upper range. Show some evidence. What I am wondering about when I am reading this is that you do have such a standard. If you use WP1 and ITASE for your slope/interscept the RICE will be the middle check standard also falling very close to the measurements' level. So the data is there. Adjust you calibration scheme to use WP1 and ITASE then use RICE as check.

Thank you for this suggestion. This paragraph was perhaps poorly phrased. Referring again to accepted laboratory guidelines and the principle of identical treatment, working standards should be similar in isotopic value to the samples being analysed (Werner and Brand 2001). We have followed these guidelines in selecting RICE and ITASE as our calibration standards for the measurement of the ice core samples, as they bracket most of the isotopic range

found in the samples. The majority of the record covers the Holocene, where the isotopic values are close to the RICE standard. The deeper part of the core (the bottom ~100 m) is closer to ITASE and represents a colder, glacial climate. A mid-point standard (like RICE) is generally considered a good choice for drift correction. Our quality control standard, WS1, falls well above the range found in the ice core, as we have noted. Using the RICE and ITASE standards to calibrate samples in this range could be considered problematic. Our uncertainty calculation is based on the calibration of WS1 using standards that have very different isotopic values, and is thus prone to unbalanced or overestimation of error.

We emphasize that above all else, it is important that our calibration scheme be appropriate for the ice core samples. The quality-control standard is a secondary concern. Calibrating the samples using WS1 as one of the standards would not be recommended practice. While interesting, we consider testing the sensitivity of the uncertainty calculation to the selection of quality-control standards to be outside the scope of this work.

We have revised the text to clarify our reasoning:

In addition, the accepted value of WS1 is well outside the range of the RICE ice core and is much greater than the RICE and ITASE standards. In general, working standards should be similar in isotopic value to the samples being measured (Werner and Brand 2001). Ideally, we would use a quality-control standard that falls within the range of the values of our two calibration standards, RICE and ITASE. However, testing the sensitivity of the calibration error to the selection of quality-control standards is outside the scope of this manuscript.

• *"The overall system performance became more variable in 2014". Looking in fig 7 it is clear that this statement is false. Your total error has fallen considerably for 2014.*

We are not referring to the magnitude of the total error but the variation in the total from stack to stack and day to day. The new figures help to illustrate this. We have clarified this sentence in the text:

The overall system performance became more variable from day to day in 2014, despite the decrease in total error.

• *There is no single piece of evidence presented in the manuscript supporting that ice breaks, drill fluid contamination and leaks or valve degradation are reasons for poor quality data. The drill fluid would likely cause considerable effects in the form of outliers if there was a spectroscopic intereference. Have you looked into the instruments spectra and the quality of the spectral fits (fit residuals are usually saved in laser spectrometers)? All of your error*

*metrics (Allan, scatter and calibration error) are based on some assesement of the RICE and WS1 standards measurement or mq water. Thus how do breaks in the ice affect you total error? Leaks and poor pressure control can be checked by looking into the pressure log of the instrument. Do you have any indication based on these data?*

Please see our response to reviewer #2 with regard to drill fluid contamination.

A discussion of the locations and effects of breaks in the ice can be found in Pyne et al. 2018. This is referenced in the text. The breaks mainly introduce error in the depth alignment. We have discussed this in response to 2.8.

The comment that the instrument's pressure log would reveal minor leaks is puzzling. The instrument's cavity is dynamically pumped with a control valve to deliver a constant pressure. Small leaks would introduce contamination (presumably ambient air) without affecting the pressure signal suggested by the reviewer. We would expect to detect this kind of leak once it affected the slope of isotope calibrations, or was observed as a higher-than-expected vapour pressure when the vaporiser was off. On the other hand, poor performance of the pressure control valve fitted to the pump is one possible source of degraded performance that we have mentioned.

Looking more widely on this issue, our intent in the manuscript is to provide a systematic methodology for calculating uncertainty from CFA stable isotope measurements. It is not to analyse the details or particulars of the measurement campaign. While we have tried to offer reasonable explanations and educated guesses as to what might have affected our results, and some of the main sources of poor instrument performance, our point is that we can calculate uncertainty without trying to diagnose every problem that occurs during a long and complex measurement campaign.

We could not always pin down the exact reasons for poor performance, nor could we halt the measurement campaign and perform a full suite of diagnostics. One of the challenges of an ice core measurement campaign is the shear amount of samples and the complexity of the setup and components (in our case involving around 10 analytical instruments all operating simultaneously), each with their own day-to-day variations. There is a finite amount of time and resources available to complete the measurements, and often one must just keep going. There will always be unknown factors influencing instrument performance and contributing to measurement uncertainty. The framework that we provide in this manuscript is a way of approaching the problem without the need for frequent interruptions to the measurement campaign.

*3 Proposed improvements*

*Here I outline some proposed improvements and changes in methods used that the authors are welcome to consider in a new version of the manuscript.*

*3.1 Spectral estimate of the noise level*

*You can get a precision/noise estimate directly from your data. Calculate the power spectral density of a section of your data (say 10 m long). It should slowly decay reaching some sort of plateau for the very high frequencies. Take the level of this "tail" and integrade over the full range of frequencies (the surface under a straight line from $-f_{Nyquist}$ to $+f_{Nyquist}$ will be good enough.) The square root of your integral gives you a direct estimate of the noise level from you data. Do this for the full record. Be aware about the type of spectral density you calculated (single/double sided)*

Thank you for drawing our attention to this additional method of calculating precision. However, we consider such an analysis to be out of the scope of this manuscript.

*3.2 Measurement accuracy on WS1*

*The values you ontain for your accuracy based on the WS1 measurements should be presented for all your calibrations. Make a plot with all of them.*

We have added a supplement (included at the end of this document) with a whisker and box plot showing the spread of all of the calibrated WS1 measurements because we agree that it adds useful information about the variability and range of these measurements. We reiterate that the calibration error is the trueness, or error of the mean of a large set of measurements. We did not assign an accuracy to each stack based on one or two calibration cycles, as this is not statistically valid.

*3.3 The calibration stability*

*Give statistics and plots on the values of the calibration line slope and intercept. This is a very interesting statistic and it is great you have all these data. Consider switching to a scheme where WS1 and ITASE are you calibration standards and RICE is your check. Compare to the current scheme where WS1 is your check.*

We have added the statistics of the slope corrections in a supplement (included at the end of this document). While it would be interesting to test the sensitivity of the calibration error to an alternate scheme, as the reviewer suggests, we consider this to be outside the scope of this manuscript.

*3.4 A section on instrument/system differences*

*Consider a well written section on the instrumental differences between 2013 and 2014. You have a plethora of data on the instruments. Look into spectral fit quality, cavity pressure and temperature for indications of issues with these parameters. Explain better the differences between the two instruments and make sure you understand well the impact of the higher acquisition rate of the 2014 instrument.*

We have added more information on the differences between the 2013 and 2014 system. However, the purpose of this is only to clarify relevant differences that may relate to results. As explained earlier, factors like the acquisition rate are incidental (since the internal acquisition rate is the same, with the only difference being output aggregated to 1 second in 2013). Most importantly, the purpose of this paper is to describe a usable philosophy for isotope calibration that is well matched to laser spectrometry systems and the purpose of the journal.

The 2013 and 2014 setups were largely the same, but differed in the construction of the vaporizer and the delivery of the mixed vapour to the isotope analyser. In 2014, the heating element of the vaporizer was modified, and a higher sample flow was delivered directly to the IWA through an open split (Emanuelsson et al., 2015). Data was recorded at 2 Hz (0.5 s) in 2013 and at 1 Hz (1.0 s) for the remaining 260 m.in 2014. The change in 2014 was made to match the depth recording rate in both years (1 Hz). Note that this was not a change in the instrument's internal data acquisition rate, only the rate of output aggregation.

*References*

*B. D. Emanuelsson, W. T. Baisden, N. A. N. Bertler, E. D. Keller, and V. Gkinis. High-resolution continuous-flow analysis setup for water isotopic measurement from ice cores using laser spectroscopy. Atmos. Meas. Tech., 8(7):2869–2883, July 2015.*

Authors' response to RC #2

*This paper describes in some detail a method of calibrating, and uncertainty estimation, of stable water isotopes in ice cores measured by a continuous flow technique. It is described as a novel approach, but I am not sure that this claim is substantiated given that measurement of gases, chemistry and water isotopes in ice cores is now rather common, and all use somewhat similar methods of calibration, drift correction and error analysis. Perhaps it is true that the uncertainty calculation for water isotope has not been published in such a detailed manner before, and this is where the novelty lies. On the whole, the paper is well written: it reads clearly, and is logically set out, and it is interesting. I am aware that an earlier reviewer has made substantial comments about each of the error estimates, and the method of combining the errors for an overall error estimate. While I largely agree with the details of this earlier review of the manuscript, I do not find it negates the value of the paper to this journal – the discussion of how to characterise the differing contribution to the error of the measurement will be helpful to many working in the field of continuous measurements who have not yet thought through the error contributions quite as deeply as the authors here. Having worked with water isotope instruments for many years, I cannot help feeling that the total mean error estimate recorded in the abstract is qualitatively about as expected, and refining the method of obtaining it will likely not change the figures substantially. And, for example, a small change to the total error in delD of 0.74 per mille would not substantially alter the (qualitative) interpretation of the sample data, which shows a range of about 70 per mille in figure 2, though to be fair small changes in the individual isotope pair errors has a larger impact on the error on the calculated dxs. But for sure, the method by which the total error was obtained is the whole point of the paper, so I am not recommending acceptance without taking into account the criticism of my colleague with the prior review, which I am not about to repeat here in my comments, but offer some further points to address. (One of the downsides of open reviewing is that it is hard to be completely independent.)*

We understand from these comments that the method of calculating the total error and the independence of the individual error factors is the reviewer's main concern. Please see our response to reviewer #1 under section 2.2.

*Specific comments:*

*Given that this is a paper describing the calibration and error estimate, I really want to know how the local standards themselves (four are mentioned on page 4: MilliQ, RICE, WS1, ITASE) were calibrated, and how often. P4/L30 describes the local standards as the accepted value (and sometimes in the manuscript as the 'true values' – when used as a check standard). Several points here.*

*One: how were these accepted values obtained? Details of the calibration of the local standards against the primary Vienna standards (or intermediates) are needed here.*

*Two: they are described as on the VSMOW/SLAP scale at various points in the manuscript, but if I were to be pedantic, I wonder if they were actually calibrated against VSMOW-2 and VSLAP-2 since the original Vienna standards were exhausted about 10 years ago. I wonder whether the local standards here are calibrated against the primary Vienna standards, or through intermediate secondary standards – this level of detail is needed in a paper focussed on calibration.*

We will address both points one and two together. The reviewer is correct that our standards were calibrated against VSMOW-2 and SLAP-2. We apologize for this oversight. The accepted values of our working standards were obtained by independent measurement of each batch, using discrete laser absorption spectroscopy on an Isotope Water Analyzer (IWA) 35EP system in 2013, and continuous laser absorption spectroscopy with our upgraded IWA-45EP system in 2014. They were calibrated against the IAEA primary standards VSMOW-2 ($\delta^{18}O$ = 0.0 ‰, $\delta^2H$ = 0.0 ‰), SLAP-2 ($\delta^{18}O$ = -55.50 ‰, $\delta^2H$ = -427.5 ‰), and GISP ($\delta^{18}O$ = -24.76 ‰, $\delta^2H$ = -189.5 ‰) through three intermediate, secondary standards INS11 ($\delta^{18}O$ = -0.37 ‰, $\delta^2H$ = -4.2 ‰), CM1 ($\delta^{18}O$ = -16.91 ‰, $\delta^2H$ = -129.51 ‰), and SM1 ($\delta^{18}O$ = -28.79 ‰, $\delta^2H$ = -225.4 ‰). Individual aliquots were measured approximately once per week during the measurement campaigns to check for variations.

These details have been added to the manuscript:

Each batch was calibrated to the International Atomic Energy Agency (IAEA) primary standards VSMOW-2 ($\delta^{18}O$ = 0.0 ‰; $\delta^2H$ = 0.0 ‰), SLAP-2 ($\delta^{18}O$ = -55.50 ‰; $\delta^2H$ = -427.5 ‰), and GISP ($\delta^{18}O$ = -24.76 ‰; $\delta^2H$ = 189.5 ‰) using three intermediate, secondary standards INS11 ($\delta^{18}O$ = -0.37 ‰; $\delta^2H$ = -4.2 ‰), CM1 ($\delta^{18}O$ = -16.91 ‰; $\delta^2H$ = 129.51 ‰) and SM1 ($\delta^{18}O$ = -28.79 ‰; $\delta^2H$ = -225.4 ‰).

*Three: additional errors creep in when producing and maintaining the local isotope standards – each calibration step removed from the primary Vienna standard introduces a new accuracy and precision error. This error is not included here at all (it isn't there in equation 4*

*– the calibration error here, as described in section 2.4.3, means an internal error is calibrating the system, rather than the error of the actual standard). As an example, I've just checked my primary Vienna (SMOW-2 and SLAP-2), and they have uncertainties of +/- 0.02 on del18O, and +/- 0.3 on delD. My commercial standards claim uncertainties of +/- 0.2 on del18O, and +/- 1.8 on delD.*

Yes, it is true that there is some additional uncertainty from the primary standards and intermediate standards, and this error would propagate through to the total uncertainty in the usual manner. However, we believe it is negligible compared to the total. Our laboratory assumes an accumulated error in the intermediate standards of +/- 1.0 ‰ for $\delta^2H$ and +/- 0.1 ‰ for $\delta^{18}O$. The error related to the calibration of WS1 is then expressed as $\sigma^2 / (n-1)$ and added to the sum under the radical. There were 377 measurements of WS1 in 2013 and 277 measurements in 2014. This adds approximately 0.0036 ‰ ($\delta^2H$) and 3.6e-5 ‰ ($\delta^{18}O$) to the sum inside the radical and does not change the total uncertainty estimate as reported.

*Four: there is something rather worrying about the 'accepted value' of the local standards in table 1 which is not explained adequately. Why do the standards change from one year to the next? Ideally, once a local standard is prepared I bulk, and calibrated against a primary standard, the local standard is then sealed in aliquots (preferably heat sealed glass ampoules), stored refrigerated, and only opened when used.*

Please see our response to reviewer #1 (section 2.1) about the difference in local working standards from year to year. We reiterate that these are working standards and we do not expect them to remain unchanged between aliquots and batches.

It is also worth noting that calibration to primary international standards is not a unique problem to this manuscript. Standard good practice is to report both the standards used, and the accepted values used in the calibration. The reviewer's interest in this topic has reminded us to add this information. Understanding the calibration and variability within the primary standards is outside the scope of our work but undertaken by those preparing the standards (e.g. Groning et al., 2007).

*The paper mentions that the isotope values depend on the water volume in the cavity (section 2.2). This is handled by removing suspect data where the water volume has drifted away from 20,000 ppm, using a short-term averaging method, and whenever the volume fell*

*below 15,000 ppm. Just to be clear: does this remove any need to calibrate the isotope value with the water volume; is the cutting of suspect data sufficient?*

As long as the mixing ratio is relatively constant during the measurement of a stack and the two calibration cycles on either side, the water volume dependence will be corrected for (Emanuelsson et al., 2015). The target water vapour mixing ratio was 20,000 ppm throughout the measurement campaign and was monitored for sudden changes. A stable offset from this target ratio is taken into account by the calibration.

*Figure 3 and P8, L15-19. There is something really rather odd here. The manuscript (P8, L15-19) claims that the drift correction might be 'poorly adapted to the upper range', and that 'Ideally, we would use a quality-control standard that falls between the two calibration standards, RICE and ITASE.' Yet, there are four local standards available: RICE and ITASE, plus MilliQ and WS1. I see no reason why the 'calibration standards' could not have been WS1 and ITASE (or even MilliQ and ITASE) which would have encompassed the whole range of the samples (figure 3 shows that samples fall above the RICE standard for example), and given one (or two) check standards that are within standard range.*

Please see our response to reviewer #1 (section 2.9). We have rewritten this paragraph. We reiterate that our calibration scheme was designed to be appropriate for the ice core samples, not our quality-control standard WS1. Although the ice core samples do on occasion fall slightly above the RICE standard, as visible in Fig. 3, this is small compared to the difference between RICE and WS1. The data in this figure comes from the upper portion of the core, representing the Holocene. Deeper in the core, corresponding to colder, glacial periods, the isotopic values are much lower (closer to ITASE) and even further from WS1. The RICE standard was also used for drift correction. In this respect our calibration was efficient to carry out. While it is true that the range between WS1 and ITASE encompassed everything, we chose RICE and ITASE as the most appropriate for our samples and reserved the others (WS1 and Milli-Q) as a check or backup if needed. Given that our calibrations as originally performed seem reasonable, we think the benefits of changing the scheme would be negligible.

*Figure 3. Could the legend box be moved off the actual data? Currently, figure 3 amply demonstrates the system drift – all standards are drifting to lighter isotope ratio. But the legend box obliterates the early ITASE standard.*

Yes, the position of the legend in Fig. 3 has been modified so that the ITASE data is visible.

*Figure 4. The upper panel is poorly produced, with no explanation about the different grey curves, or why one Allan plot does not curve upwards. The dashed 'discrete precision' line is grey in the legend, and black in the figure. The lower panel shows the Allan error parabola for each stack – it would clarify what is demonstrated here to explain that the lowest part of the dip is the start/end of the stack of ice, and the point at which the calibration is carried out.*

The upper panel was meant as a conceptual diagram, but it is evident that this was not clear from the text, and the details of the curves are irrelevant to the present analysis (see response to reviewer #1). We have removed it. We now show the Allan error for the entire range of depths and have noted in the caption that the dips are the start and end of a stack:

(caption) Figure 4: Allan error vs. depth, in per mil. $\delta^2$H is in blue and $\delta^{18}$O is in red. The low points of the dips are the start and end of a stack, between which calibrations were carried out.

*Abstract and P8, L25-30, Summary. Have you any evidence that drill fluid is influencing the error? What fluid is being used? What is its vapour pressure? Could it actually build up in the instrument, or would it simply evaporate at the low cavity pressure and high temperature and be swept out with the waste gas?*

The drill fluid is a combination of Estisol-240 and Coasol (Bertler et al., 2018). We have noted this in the revised text. Estisol-240 and Coasol are organic compounds which we have tentatively identified as an ester and diester with a vapor pressures < 0.01 kPa at 20°C. However, vapor pressures may become more relevant at temperatures in the vaporiser chamber and heated lines entering the laser spectrometer. As noted earlier, the purpose of this paper is not to speculate on or determine the level of contamination, but rather to explain the design of the data analysis system that is robust to a reasonable extent of issues with unknown interfering contaminants in the spectrometer system.

To further outline what we understand about the drill fluid contaminants, but consider outside the scope of what should be published, we explain further here. The drill fluid contaminants formed visible accumulations through the system into the vaporiser. Our GC-MS analysis of the contaminant (as recovered from fresh samples and the the drill hole) revealed strong matches to branched ester structures (Estisol-240) and diester structures (Coasol). We did not however attempt to analyse contamination in the vaporiser or analyser.

The drill fluid itself is not expected to have absorbances likely to interfere with water isotope absorption lines at approximately 1.4 μm (or 3663 cm-1), but this is difficult to verify. Interferences in this wavelength range are more likely from organic compounds with hydroxyl groups. Unfortunately, at the temperatures in the vaporiser (170°C), it is reasonable to expect that the esters and diesters would hydrolyse to form compounds with hydroxyl groups. Based on the most likely structures we have identified for Estisol-240 and Coasol, the respective hydrolysis products appear likely to include compounds such as 2-ethyl hexanoic acid, 2-ethyl hexanol, 2-methyl propanol, and pentanedioic acid. Of these compounds, the latter three have strong absorbance features in the 3663 cm-1 region, and the latter has strong absorbance features between 3550 and 3600 cm-1. Our best guess is that small amounts of these degradation products, or the original esters, can reach the laser cavity. There, it seems reasonable to expect quite variable gas phase interferences, or more likely, deposits on cavity mirrors that cause slow degradation of instrument performance.

Thus, we conclude that there is a very reasonable case for interferences that would be difficult to quantify because they are derived from degradation products of the main contaminants. We argue that this case is somewhat too speculative to include in the published work, and not of great relevance to the journal because scientists carrying out measurement techniques should generally work to prevent contamination rather than quantify and characterise its chemistry. Our analysis is strongly supportive of the level of commentary we have included in the manuscript, namely that degraded analytical performance can be considered a realistic feature of a melting campaign such as we carried out.

*Are system leaks and valve degradation really affecting the data? What evidence is there, or is this speculation? (I appreciate the text does say 'could affect. . ..').*

Our evidence is mostly anecdotal. When we did tighten the valves, the noise in the data was reduced for a time, but gradually increased to previous levels (this variation is visible in the new Fig. 5). We reiterate our view that during the time pressures of the melt campaign, when

many instruments and researchers were assembled, there was insufficient time to pursue efforts to definitively isolate and identify problems. Instead, data quality met the criteria deemed acceptable to proceed, and as a result the opportunity to definitively clarify whether degraded performance was related to drill fluid contamination, minor leaks (that did not alter cavity pressure), or another unidentified cause.

*Following this last point, figures 5 and 7 don't help me understand the scatter and total error over the full record. The scatter error is greater in 2014, due to poor core quality, build up of drill fluid and system leaks – yes? But the overall error is better in 2014 (figure 7). What is surely need here to help clarify this point is a scatter error figure that encompasses the whole core, rather than the short 2013 section in figure 5.*

Figure 5 has been modified to show the whole record. We hope this resolves the issue.

*Perhaps best of all would be a single figure that for the whole core demonstrates the total error and the three component errors – is this possible?*

Yes, all of the error factors have been added to Figure 7, which also shows the total error.

Additional References:

Analytical Methods Committee, Technical Brief, No. 13, Terminology - the key to understanding analytical science. Part 1: Accuracy, precision and uncertainty, Royal Society of Chemistry, September 2003.

Brand, W.A., Coplen, T.B., Aerts-Bijma, A.T., Böhlke, J.K., Gehre, M., Geilmann, H., Gröning, M., Jansen, H.G., Meijer, H.A., Mroczkowski, S.J. and Qi, H.: Comprehensive inter-laboratory calibration of reference materials for δ18O versus VSMOW using various on-line high-temperature conversion techniques. *Rapid Comm. Mass Spectrom.*, *23*, 999-1019, 2009.

Carter, J. F. and Fry, B.: Ensuring the reliability of stable isotope ratio data—beyond the principle of identical treatment, Anal. Bioanal. Chem., 405, 2799-2814, 2013.

Groning, M., Van Duren, M., and Andreescu, L.: Metrological characteristics of the conventional measurement scales for hydrogen and oxygen stable isotope amount ratios: the δ-scales. In: Combining and Reporting Analytical Results, Eds. A. Fajgelj, M. Belli, U. Sansone. Proceedings of an International Workshop on "Combining and reporting analytical results: The role of traceability and uncertainty for comparing analytical results", Rome, 6-8 March 2006. Royal Society of Chemistry (2007) 62-72.

Kirchner, J.: Data Analysis Toolkits: http://seismo.berkeley.edu/~kirchner/eps_120/Toolkits/Toolkit_05.pdf, 2001 (last access: 31 May 2018).

Meier-Augenstein, W.: *Stable isotope forensics: an introduction to the forensic application of stable isotope analysis*. Vol. 3. John Wiley & Sons, 2011.

Munksgaard, N.C., Cheesman, A.W., Gray-Spence, A., Cernusak, L.A., and Bird, M.I.: Automated calibration of laser spectrometer measurements of δ18O and δ2H values in water vapour using a Dew Point Generator, Rapid Commun Mass Spectrom, 32, 1008–1014. https://doi.org/10.1002/rcm.8131, 2018.

Paul, D., Skrzypek, G. and Fórizs, I.: Normalization of measured stable isotopic compositions to isotope reference scales – a review. Rapid Commun. Mass Spectrom., 21, 3006-3014. doi:10.1002/rcm.3185, 2007.

Warming, E., Svensson, A., Vallelonga, P. and Bigler, M.: A technique for continuous detection of drill liquid in ice cores. J. Glaciol., 59, 503-506, 2013.

Wassenaar, L.I., Hendry, M.J., Chostner, V.L. and Lis, G.P.: High resolution pore water δ2H and δ18O measurements by H2O (liquid)– H2O (vapor) equilibration laser spectroscopy. Environ. Sci. Technol., 42, 9262-9267, 2008.

Werner, R.A. and Brand, W.A.: Referencing strategies and techniques in stable isotope ratio analysis. Rapid Comm. Mass Spectrom., 15, 501-519, 2001

**Revised Figures:**

[Figure]

**Figure 1: An example of the raw data from a full day of ice melting and calibration cycles (2-3 July 2014): (a) δ²H, (b) δ¹⁸O, and (c) water vapour mixing ratio. Isotope data that were removed because of water concentration anomalies are marked in red in (a) and (b) panels.**

[Figure]

**Figure 2: A selected example section of δ²H vs. depth. The data marked in red represent the transitions between the Milli Q standard and ice core at the boundaries of each 3-metre stack. These data points (and other poor quality data) were removed from the final dataset.**

[Figure]

**Figure 3: Time vs. raw δ¹⁸O (uncalibrated) for one day of melting (3 July 2014). Values of standards drift noticeably over the course of the day. An example of one calibration cycle of three water standards run between ice core stacks are marked in colour: WS1 (red), RICE (green), and ITASE (blue).**

[Figure]

**Figure 4: Allan error vs. depth, in per mil. δ²H is in blue and δ¹⁸O is in red. The low points of the dips are the start and end of a stack, between which calibrations were carried out.**

[Figure]

Figure 5: Scatter error vs. depth, in per mil. $\delta^2$H is in blue and $\delta^{18}$O is in red.

[Figure]

**Figure 6: Representative δ¹⁸O calibration of ice core stack and WS1, using RICE and ITASE standards from the same cycle, 15-second moving average vs. time (measured on 2 Jul 2014). The difference between the "true" value of WS1 (blue) and the calibrated measured value of WS1 (red) is the calibration error. The error that was applied to the CFA dataset is the average difference of all WS1 calibration measurements during the melting campaign.**

[Figure]

**Figure 7: Total uncertainty vs. depth, along with each individual error factor, in per mil. Top: δ²H. Bottom: δ¹⁸O. There is a noticeable discontinuity at 500 m; the melting campaign was paused at 500 m in 2013, and melting was resumed in 2014 with a modified setup. The reduced calibration error in 2014 is responsible for the large step down in total error.**

**Supplementary Material**

**Figures**

[Figure]

**Figure S2. Box and Whisker plot showing the spread of WS1 δ¹⁸O (left) and δ²H (right) measurements (after calibration using Eq. 3) used to calculate the calibration error. The median is shown as a red line. The blue boxes indicate the 25th and 75th percentiles. Red crosses indicate outliers. Green star indicates the true value of WS1 as determined in each year.**

[Figure]

**Figure S2. Box and Whisker plot showing the range of calculated slopes (Eq. 2) used to calibrate the δ¹⁸O (left) and δ²H (right) isotope data in Eq. 3. The median is shown as a red line. The blue boxes indicate the 25th and 75th percentiles. Red crosses indicate outliers.**

[Figure]

**Figure S3. Box and Whisker plot showing the range of RICE standard δ¹⁸O (left) and δ²H (right) measurements (raw) used to calibrate the isotope data in Eq. 3. The median is shown as a red line. The blue boxes indicate the 25ᵗʰ and 75ᵗʰ percentiles. Red crosses indicate outliers. Green star indicates the true value of RICE as determined in each year.**

**Tables**

**Table S3. Mean and standard deviation (SD) for WS1 after calibration (using Eq. 3), and RICE (raw measurements) and slope corrections used in Eq. 3 to calibrate the isotope data.**

[revised manuscript text omitted]

$$\sim\!\!\!\!slope\, m_i = \frac{\delta^T_{RICE}\sim\!\!RICE_{true} - \delta^T_{ITASE}\sim\!\!ITASE_{true}}{\delta_{RICEi}\sim\!\!RICE_i - \delta_{ITASEi}\sim\!\!ITASE_i}$$
( 2 )

Where $\delta^T_{RICE}\sim\!\!RICE_{true}$ and $\delta^T_{ITASE}\sim\!\!ITASE_{true}$ are the accepted standard values and $\delta_{RICEi}\sim\!\!RICE_i$ and $\delta_{ITASEi}\sim\!\!ITASE_i$ are the *ith* measured value of the standards RICE and ITASE, respectively. The correction then takes the following form:

$$\delta_{corrected} = \frac{\delta^T_{RICE} - \delta^T_{ITASE}}{\delta_{RICEi} - \delta_{ITASEi}} * (\delta_{raw} - \delta_{RICEi}) + \delta^T_{RICE}$$
(3)

By design, the y-intercept $b$ is $\delta^T_{RICE}$. We calculated this correction for each stack using the closest set of RICE and ITASE calibration measurements both before and after the stack. We then apply  the  correction to each data point , weighting the factors calculated from the calibrations before and after the stack by the time difference between the data point and the calibration:

$$\delta_{corrected}(t) = [(\delta_{raw}\sim\!\!\delta - \delta_{RICE1}\sim\!\!RICE_1) * \sim\!\!slope\, m_1 + \delta^T_{RICE}\sim\!\!RICE_{true}] * (1 - f)\sim\!\!(1 - t) * step + [(\delta_{raw}\sim\!\!\delta - \delta_{RICE2}\sim\!\!RICE_2) * \sim\!\!slope\, m_2 + \delta^T_{RICE}\sim\!\!RICE_{true}] * \sim\!\!t * step\, f$$
(  4 )

where $\delta_{raw}\sim\!\!\delta_i$ is the uncalibrated raw δ²H or δ¹⁸O value of the  ice core sample, $\delta_{RICE1}\sim\!\!RICE_1$ and $\delta_{RICE2}\sim\!\!RICE_2$ are the measured values of the RICE standard before and after the stack, respectively, $t$ is the time of $\delta_{raw}\sim\!\!\delta$ measurement relative to $t_1$,  and $f$ is a dimensionless weighting factor:  $f = \sim\!\!(t_2 - t_1)^{-1} \, t/(t_2 - t_1)$, $t_1$ = starting time of $\delta_{RICE1}\sim\!\!RICE_1$  measurement before the stack, and $t_2$ = ending time of $\delta_{RICE2}\sim\!\!RICE_2$  measurement after the stack. We note that this method assumes that drift is approximately linear over the measurement period. Our calibration procedure was validated by comparison to discrete measurements in Emanuelsson et al. (2015). Figures showing the  range of slope corrections and the RICE standard measurements used to calibrate the data in each year can be found in the Supplementary Material. Mean values and standard deviations are in Table S1.

**2.4 Uncertainty calculation**

We identified three main sources of  uncertainty in our measurements: (i) the Allan variance error (a measure of our ability to correct for drift, a systematic source of uncertainty due to instrumental instability ), (ii) the scatter or

noise in the data over  our chosen averaging interval, and (iii) a general calibration error relating to the overall accuracy of our calibration . Our three error factors can be formally categorized as follows:

1. "Scatter error": error of the variance / precision / random variation of replicate measurements
2. "Calibration error": error of the mean / trueness
3. "Allan variance error": systematic error or bias due to our imperfect ability to correct for drift

The first two can be quantified with general analytical expressions (Kirchner, 2001). Systematic error or bias does not have a general analytical form; isotopic drift is fortunately amenable to correction, but the method is imperfect.

We assume that the three error factors are uncorrelated to a large degree. This is supported by the general framework that we have used (Kirchner, 2001; Analytical Methods Committee, 2003) and the actual errors calculated at each data point ($R^2 <$ 0.05 in each year for both isotopes). In practice it is impossible for all error factors to be completely uncorrelated, as some underlying sources of error will affect all aspects of the system. However, we believe these interactions to be small and/or short-lived and negligible to the total uncertainty. With this assumption, We calculate each  error factor separately and add them in quadrature to arrive at the total uncertainty estimate:

$$\sigma\epsilon_{total} = \sqrt{\sigma\epsilon^2_{allan~AVlE} + \sigma\epsilon^2_{scatter} + \sigma\epsilon^2_{calib}}$$

( 5 )

[revised manuscript text omitted]

30    $$\epsilon_{scatter} = mean(\sigma_i/\sqrt{n_i}), \ i = 1 \dots N \ \sigma_{scatter} = \frac{1}{N}\sum_1^N \frac{\sigma_i}{\sqrt{n_i}} = mean(\sigma_i/\sqrt{n_i}), \ i = [1 \dots N]$$
(  9 )

where $\sigma_i$ is the standard deviation, $N$ is the total number of intervals, and $n_i$ is the number of data points in the $ith$ interval ($n = $ ~30 in 2013 and ~15 in 2014). We note that the number of points that are contained in the interval is different in 2013 and

2014, resulting from the difference in output aggregation (not the instrument's internal data acquisition rate). This could affect the amount of noise in the data. However, we have not attempted to analyse this in detail, as we are only concerned here with quantifying the uncertainty associated with our averaging interval, regardless of the number of data points averaged.

[revised manuscript text omitted]

Among the three error factors, the general calibration error is the largest contributor to the total error in 2013: $\epsilon\sigma_{calib}$ ($\delta^2$H) = 0.80 ‰ and $\sigma\epsilon_{calib}$ ($\delta^{18}$O) = 0.12 ‰. However, this error is greatly reduced for 2014: $\epsilon\sigma_{calib}$ ($\delta^2$H) = 0.22 ‰ and $\epsilon\sigma_{calib}$ ($\delta^{18}$O) = 0.078 ‰, reflecting the improved  measurement of the "true" standard values.

15   We e were not able to measure the standards against VSMOW/SLAP using the 2013 CFA setup, which would provide a better comparison between measured and accepted values, following from the principle of identical treatment (Werner and Brand 2001). The 2013 $\sigma\epsilon_{calib}$ is thus likely to be a very conservative estimate of the error. In addition, the accepted value of WS1 is well

20  outside the range of the RICE ice core and is much greater than the RICE and ITASE standards. In general, working standards should be similar in isotopic value to the samples being measured (Werner and Brand 2001),  
[revised manuscript text omitted]

[Figure]

[Figure]

[Figure]

**Figure 4:**  **Allan error vs. depth**, in per mil  $\delta^2 H$ is in blue and $\delta^{18} O$ is in red. The low points of the dips are the start and end of a stack, between which calibrations were carried out.

[Figure]

[Figure]

**Figure 5: Scatter**  **error vs. depth**, in per mil . $\delta^2H$ is in blue and $\delta^{18}O$ is in red.

[Figure]

[Figure]

[Figure]

**Figure 6: Representative δ¹⁸O calibration of ice core stack and WS1, using RICE and ITASE standards from the same cycle, 15-second moving average vs. time (measured on 2 Jul 2014). The difference between the "true" value of WS1 (blue) and the calibrated measured value of WS1 (red) is the calibration error. The error that was applied to the CFA dataset is the average difference of all WS1 calibration measurements during the melting campaign.**

[Figure]

[Figure]

**Figure 7: Total**  uncertainty **vs. depth****, along with each individual error factor, in per mil****.** **Top: δ²H. Bottom: δ¹⁸O. There is a noticeable discontinuity at 500 m; t****T**he melting campaign was **paused at 500 m** in 2013**, and melting was resumed in 2014 with a modified setup. The reduced calibration error in 2014 is responsible for the large step down in total error.**

**Tables**

Table 1: Accepted values (VSMOW/SLAP scale) for water standards used for calibrations, in per mil (‰).

| | δ¹⁸O | | δ²H | |
|---|---|---|---|---|
| **Water standard** | **2013** ‰ | **2014** ‰ | **2013** ‰ | **2014** ‰ |
| Milli Q | -5.89 | n/a | -34.85 | n/a |
| WS1 | -10.84 +/- 0.10 | -10.83 | -74.15 +/- 0.94 | -74.85 |
| RICE | -22.54 +/- 0.05 | -22.27 | -175.02 +/- 0.19 | -173.06 |
| ITASE | -37.39 +/- 0.05 | -36.91 | -299.66 +/- 0.18 | -295.49 |

Table 2: Summary of error estimates, in per mil (‰).

| | δ¹⁸O | | | δ²H | | |
|---|---|---|---|---|---|---|
| **Error factor** | **2013** +/- ‰ | **2014** +/- ‰ | **Combined** +/- ‰ | **2013** +/- ‰ | **2014** +/- ‰ | **Combined** +/- ‰ |
| Allan | 0.16 | 0.11 | 0.14 | 0.13 | 0.083 | 0.12 |
| Scatter | 0.093 | 0.13 | 0.10 | 0.26 | 0.37 | 0.29 |
| Calibration | 0.12 | 0.078 | n/a | 0.80 | 0.22 | n/a |
| **Total** | **0.22** | **0.19** | **0.21** | **0.85** | **0.43** | **0.76** |